# FideDiff: Efficient Diffusion Model for High-Fidelity Image Motion Deblurring

**Xiaoyang Liu**[1,*], **Zhengyan Zhou**[1,*], **Zihang Xu**[1],
**Jiezhang Cao**[1,2], **Zheng Chen**[1], **Yulun Zhang**[1,†]
[1]Shanghai Jiao Tong University, [2]Harvard University

## Abstract

Recent advancements in image motion deblurring, driven by CNNs and transformers, have made significant progress. Large-scale pre-trained diffusion models, which are rich in real-world modeling, have shown great promise for high-quality image restoration tasks such as deblurring, demonstrating stronger generative capabilities than CNN and transformer-based methods. However, challenges such as unbearable inference time and compromised fidelity still limit the full potential of the diffusion models. To address this, we introduce FideDiff, a novel single-step diffusion model designed for high-fidelity deblurring. We reformulate motion deblurring as a diffusion-like process where each timestep represents a progressively blurred image, and we train a consistency model that aligns all timesteps to the same clean image. By reconstructing training data with matched blur trajectories, the model learns temporal consistency, enabling accurate one-step deblurring. We further enhance model performance by integrating Kernel ControlNet for blur kernel estimation and introducing adaptive timestep prediction. Our model achieves superior performance on full-reference metrics, surpassing previous diffusion-based methods and matching the performance of other state-of-the-art models. FideDiff offers a new direction for applying pre-trained diffusion models to high-fidelity image restoration tasks, establishing a robust baseline for further advancing diffusion models in real-world industrial applications. Our dataset and code will be available at `https://github.com/xyLiu339/FideDiff`.

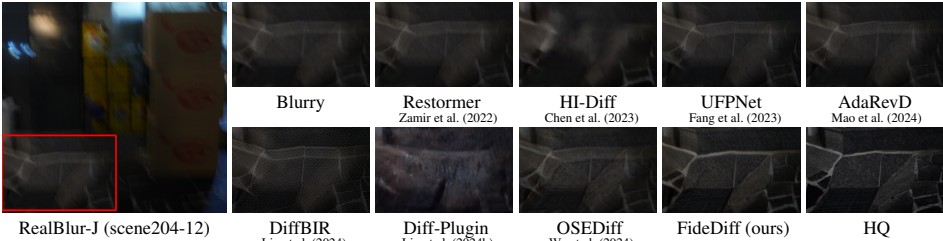

Figure 1: Previous models often suffer from the real-world generalization.

## 1 Introduction

Image motion deblurring has long been considered an ill-posed image restoration problem due to the complex formation causes, such as camera shake and high-speed object motion during exposure time. In recent years, with the development of CNNs and transformers, significant progress has been made in the field of motion deblurring, including many end-to-end deblurring methods (Mao et al., 2024; Cheng et al., 2025) and motion blur kernel estimation-based approaches (Kim et al., 2024). However, these specialized models often lack a true understanding of the world, which limits their ability to handle unknown situations and generalize to real-world scenarios, as shown in Fig. 1.

In recent years, large-scale pre-trained diffusion models (DMs) have been gradually employed for image restoration tasks, showing remarkable generalization and high-quality restoration ability, bringing new hope to the deblurring field (Lin et al., 2024; Liu et al., 2024b). However, similar to the development of diffusion in many low-level vision tasks, several challenges hinder the full utilization of large-scale DMs, including a) unbearable inference time, usually with tens or hundreds of sampling steps (Lin et al., 2024), and b) struggling with fidelity (Liu et al., 2024b) or even overlooking it to emphasize the perceptual quality (Cheng et al., 2025).

---

[*]Equal contribution
[†]Corresponding author: Yulun Zhang, yulun100@gmail.com

Additionally, there appears to be a trade-off (Fig. 2) between the inference steps (time) and fidelity for DMs. Take well-developed super-resolution as an example, many few- or one-step approaches struggle to maintain fidelity and sacrifice it to pursue the overall quality. This shift undermines the core objective of image restoration, which is to restore an image to its original form, and deviates from practical industry-oriented tasks.

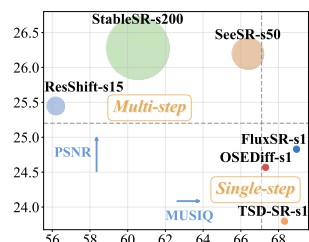

Figure 2: Fidelity-perception tradeoff w.r.t. steps in SR task.

To address these challenges and better harness the potential of pre-trained DMs for efficient and high-fidelity image motion deblurring, we introduce our model, the high-**Fi**delity single-step **de**blurring **Diff**usion model, **FideDiff**, which reduces the sampling steps to a single iteration while prioritizing restoration fidelity. Unlike previous single-step diffusion approaches (Wu et al., 2024; Dong et al., 2025; Li et al., 2025a) that assign a fixed timestep to all low-quality images, we reformulate motion deblurring as a diffusion-like process where each timestep corresponds to a specific blur severity. We analyze the challenges of traditional modeling and define the forward process through blur trajectories, then directly optimize a consistency objective that forces all timesteps to predict the same clean image. To enable this, we reconstruct the GoPro (Nah et al., 2017) dataset to provide matched blur trajectories, allowing the model to learn cross-time consistency and naturally support accurate one-step deblurring during inference.

We carefully build a foundation model with pretrained diffusion priors for high-fidelity deblurring. Considering that effective utilization of the blur kernel and incorporating control information into diffusion-based deblurring methods are important but less explored, we propose Kernel ControlNet, which introduces blur kernel estimation and cleverly integrates control information in the form of filters into the trained foundation model, further improving deblurring performance. We also design a regression module for timestep prediction, enabling the model to adaptively select the appropriate timestep based on blur degree during inference, allowing it to handle various scenarios more flexibly.

Our contributions are as follows: a) We reformulate the diffusion process in deblurring and propose a time-consistency training paradigm to support one-step sampling. b) We develop a robust single-step high-fidelity foundation model, and c) introduce Kernel ControlNet to enhance performance, along with a t-prediction module for dynamic blur level prediction. Our model outperforms all previous pre-trained diffusion-based models on full-reference metrics, and it matches or even surpasses the well-developed transformer-based models on perceptual similarity (LPIPS/DISTS). Our work provides a new perspective for applying pre-trained DMs to image restoration tasks and establishes a robust baseline, which can further refine and expand the application of DMs in low-level vision, facilitating their better application in real-world industrial scenarios.

## 2 RELATED WORKS

### 2.1 IMAGE MOTION DEBLURRING METHODS

Image motion deblurring has been widely studied, from traditional handcrafted models to deep learning approaches, with early methods framing it as an inverse problem, using priors like total variation (Chan & Wong, 1998), hyper-Laplacian (Krishnan & Fergus, 2009), or gradient statistics (Shan et al., 2008; Xu et al., 2013) to regularize deconvolution. Recently, CNN and transformer-based methods have dominated, including many end-to-end methods (Zamir et al., 2022; Chen et al., 2022; Kong et al., 2023), facilitating downstream visual understanding such as Hu et al. (2023); Liu et al. (2025b). Park et al. (2020) progressively removes blur with a multi-temporal recurrent neural network. Mao et al. (2024) uses an adaptive patch exiting reversible decoder, while Liu et al. (2024a) introduces a motion-adaptive separable collaborative filter to handle complex motion blur.

### 2.2 KERNEL ESTIMATION

Kernel estimation plays an important role in both blind and non-blind deblurring. Previous methods assume either uniform blur (Sanghvi et al., 2024; Zhang et al., 2024), which limits real-world applicability, or predict pixel-wise kernels (Sun et al., 2015; Carbajal et al., 2021; Gong et al., 2017) for subsequent non-blind deblurring (Dong et al., 2020; Tai et al., 2011; Tang et al., 2023). Fang et al. (2023) represents motion blur kernels in a latent space with normalizing flows and uses uncertainty learning to improve end-to-end model performance. Kim et al. (2024) presents an efficient deblurring model that decomposes the task into blur pixel classification and discrete-to-continuous conversion, achieving high performance with low computation cost.

## 2.3 DIFFUSION MODELS

Diffusion Models (DMs) (Rombach et al., 2022; Esser et al., 2024; Labs, 2024) have recently made significant progress in the generation field, and become powerful tools in many low-level vision tasks such as real-world super-resolution (Wu et al., 2024; Li et al., 2025a) and image restoration tasks, such as deblurring (Cheng et al., 2025; Liu et al., 2025a) and many other tasks (Wang et al., 2025; Guo et al., 2025; Gong et al., 2025). In the deblurring field, Xia et al. (2023) and Chen et al. (2023) use additional diffusion architecture to construct conditional priors to strengthen the network. Meanwhile, Whang et al. (2022) and Ren et al. (2023) apply diffusion in the image space to denoise blurry images. The deblurring methods based on pretrained DMs, UID-Diff (Cheng et al., 2025) and Diff-Plugin (Liu et al., 2024b), leverage unsupervised learning with unpaired data and the blur information plugin approach, respectively, to tame pretrained DMs to learn the deblurring.

However, diffusion-based methods often have high inference cost and require strong priors. And the trade-off between fidelity and perceptual realism of pretrained DMs is non-trivial. Some diffusion outputs may look perceptually plausible but deviate from ground truth.

## 3 TASK FORMULATION AND DATA PREPARATION

### 3.1 MOTIVATION

Despite the advances in accelerating diffusion processes, the application of one-step diffusion to image restoration has not been sufficiently studied. Distillation-based paradigms dominate existing efforts, for example, SinSR (Wang et al., 2024) and FluxSR (Li et al., 2025a) compress multi-step teachers into single-step students, while OSEDiff (Wu et al., 2024) and TSD-SR (Dong et al., 2025) adopt score or distribution distillation to regress toward sharp images. While effective at reducing inference cost, such approaches suffer from two fundamental limitations.

1. Loss of Diffusion's Inductive Bias: Collapsing the iterative denoising into a single mapping with a fixed timestep reduces diffusion to a direct regression model, resembling a Unet with strong initialization but detached from diffusion principles. Such an application is crude without proper modeling and, intuitively, is unsuitable for handling different levels of degradation, such as deblurring. Ignoring this inductive bias risks sacrificing diffusion's robustness and fidelity.
2. Target Inconsistency in Image Restoration: Super-resolution and motion deblurring are full-reference tasks aimed at recovering images close to the clean target. However, most one-step methods prioritize no-reference perceptual metrics (e.g., CLIPIQA (Wang et al., 2023), MUSIQ (Ke et al., 2021)), sacrificing full-reference fidelity (e.g., PSNR, LPIPS (Zhang et al., 2018)), as shown in Fig. 2. Pretrained generative priors (e.g., Stable Diffusion (Rombach et al., 2022; Esser et al., 2024)) encourage models to generate perceptually pleasing but less faithful reconstructions, inflating perceptual scores at the cost of restoration goals.

### 3.2 PRELIMINARY

According to the standard diffusion process (Ho et al., 2020; Rombach et al., 2022), we define

$$q(z_{1:T}|z_0) = \prod_{t=1}^{T} q(z_t|z_{t-1}), \quad q(z_t|z_{t-1}) = \mathcal{N}(z_t; \sqrt{1-\beta_t}z_{t-1}, \beta_t I), \tag{1}$$

where $z_t$ is the latent variable at time step $t$, which is progressively corrupted by Gaussian noise starting from $z_0$. The mean and variance are determined by the noise schedule $\beta_t$.

In the reverse process, the mean value of the Gaussian distribution $q(z_{t-1}|z_t, z_0)$ is calculated as:

$$\mu_t(z_t, z_0) = \frac{\sqrt{\alpha_t}(1-\bar{\alpha}_{t-1})}{1-\bar{\alpha}_t}z_t + \frac{\sqrt{\bar{\alpha}_{t-1}}\beta_t}{1-\bar{\alpha}_t}z_0, \tag{2}$$

where $\alpha_t = 1 - \beta_t$ and $\bar{\alpha}_t = \prod_{i=1}^{t} \alpha_i$. The denoising network $\epsilon_\theta$, for predicting $p_\theta(z_{t-1}|z_t)$, needs to minimize the difference with $q(z_{t-1}|z_t, z_0)$. Estimated $\hat{\epsilon}$ from $\epsilon_\theta$ is used to generate the $\hat{z}_0$ via

$$\hat{z}_0 = \frac{z_t - \sqrt{1-\bar{\alpha}_t}\hat{\epsilon}}{\sqrt{\bar{\alpha}_t}}, \quad \hat{\epsilon} = \epsilon_\theta(z_t, c, t). \tag{3}$$

Here, $c$ represents the control information, like text, image, or other conditions. With Eq. 3, at each step, $\epsilon_\theta$ can directly restore the clean $z_0$. However, for stable and high-quality generation, denoising from $z_{t-1}$ is preferred over directly exiting the backward process, as seen from Eq. 2.

### 3.3 FORWARD AND BACKWARD REFORMULATION FOR DEBLURRING

Motion blur in images is caused by the relative motion between the camera and the scene during the exposure time. When capturing an image, each point in the scene is projected onto the image plane over time. This can be mathematically represented as an integral over the exposure time, and alternatively, it can also be approximated by a pixel-wise convolution, where the blur kernel $K$ represents the motion path of the camera or object, and the blurred image $I_{\text{blur}}$ is obtained by convolving the sharp image $I_{\text{sharp}}$ with the blur kernel $K$ and adding noise $n$:

$$I_{\text{blur}} = \frac{1}{t_1 - t_0} \int_{t_0}^{t_1} Sensor(t)\, dt \approx I_{\text{sharp}} * K + n. \tag{4}$$

In Fig. 3, we reformulate the forward and backward processes for the image motion blurring and deblurring. We define the clean image as $z_0$ and the initial blur kernel as identity convolution $k_0$, where $z_0 = z_0 * k_0$. From a pure clean image to the blurry image, we regard the forward blur kernel generation process as a chain, following:

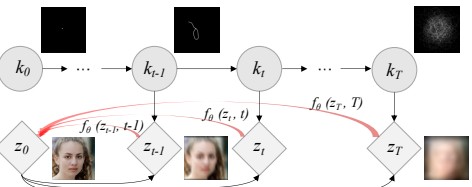

Figure 3: Forward and backward processes.

$$q(k_{1:T}|k_0) = \prod_{t=1}^{T} q(k_t|k_{t-1:0}). \tag{5}$$

Commonly used kernel generation methods involve generating random blur trajectories, which are convolved with sharp images to produce blurry counterparts. In these methods, the blur kernel typically exhibits non-linear and non-uniform trajectories, with $k_t$ depending on previous states $k_{t-1:0}$ based on different simulation techniques. For simplicity, Figure 3 illustrates a globally uniform blur, while real-world scenarios apply the blur kernel on a pixel-wise basis.

At each state in the forward process, $z_t$ is calculated as $z_t = z_0 * k_t$. For backward process, $q(z_{t-1}|z_t)$ is intractable, so we consider $q(z_{t-1}|z_t, z_0)$ and define $q(k_t|z_t, z_0)$ as the conditional distribution of the variable $k_t$. With identity convolution $k_0$, we try to calculate $q(k_{t-1}|k_t, k_0)$ using

$$q(k_{t-1}|k_t, k_0) = q(k_t|k_{t-1}, k_0) \frac{q(k_{t-1}|k_0)}{q(k_t|k_0)}. \tag{6}$$

Once we have the distribution of $q(k_{t-1}|k_t, k_0)$, we can compute the $q(z_{t-1}|z_t, z_0)$ via $z_{t-1} = k_{t-1} * z_0$, and then optimize the probability model $\epsilon_\theta$ to minimize the KL divergence between the $p_\theta(z_{t-1}|z_t)$ and $q(z_{t-1}|z_t, z_0)$, following the approach outlined in previous work.

However, two major challenges arise. Firstly, $q(k_t|z_t, z_0)$ is generally intractable. If the blur kernel were spatially uniform across the entire image, it could be inferred directly from $z_t$ and $z_0$. In practice, however, real motion blur kernels vary pixel-wise due to depth differences, camera motion, and independently moving objects, making a deterministic solution for $k_t$ impractical. Secondly, the forward kernel generation process is highly complex and non-Markovian, as it often depends on physical factors such as point velocity, impulse, and inertia (Kupyn et al., 2018; Boracchi & Foi, 2012). Consequently, neither the marginal distribution $q(k_t|k_0)$ nor the conditional $q(k_{t-1}|k_t, k_0)$ can be accurately modeled by simple parametric distributions (e.g., Gaussians), violating the assumptions commonly exploited in standard diffusion formulations.

Despite these challenges, the problem is not unsolvable. As Eq. 2 and Eq. 3 reveal, the fundamental objective of DMs is to reconstruct $z_0$. We reformulate the training objective as a cross-time consistency regression, explicitly enforcing $f_\theta(z_t, t)$ to yield a consistent estimate of $z_0$ across all timesteps $t$. This temporal alignment naturally supports single-step inference without requiring multi-step denoising iterations. According to Schusterbauer et al. (2025) and Tong et al. (2024), the need for multiple sampling steps in standard diffusion arises primarily from the random pairing between Gaussian noise samples and data points during training. If the blur trajectory of each image is known, and all points along the backward trajectory are jointly trained to map toward the same clean target, the model learns an intrinsic temporal consistency. Formally, enforcing

$$z_0 = f_\theta(z_t, t) = f_\theta(z_{t'}, t'), \quad \min_\theta \mathbb{E}_{t, z_0} \big\| f_\theta(z_t, t) - z_0 \big\|^2 \tag{7}$$

promotes trajectory consistency (Song et al., 2023) and facilitates accurate one-step sampling.

| Avg. Frames | 5 | 7 | 9 | 11 | 13 | 15 | Total |
|---|---|---|---|---|---|---|---|
| GoPro | 0 | 175 | 0 | 1,818 | 110 | 0 | 2,103 |
| GoPro (Enlarged) | 838 | 980 | 869 | 2,081 | 1,900 | 1,209 | 7,877 |

Table 1: Statistics of GoPro image training pairs.

In this work, we base our consistency training on the pretrained Stable Diffusion model. As previously mentioned, DMs directly target $z_0$. Among different optimization objectives (e.g., $\mu$-, $\epsilon$-, or $v$-prediction (Salimans & Ho, 2022)), we opt for the Stable Diffusion 2.1 base model with $\epsilon$-prediction for simplicity, which is commonly used in related tasks. To better adapt the pretrained DMs to the deblurring task, we retain the original diffusion coefficients $\alpha_t$ and $\beta_t$, and adjust the $\hat{\epsilon} = \epsilon_\theta$ such that the following equation holds, where $\hat{\epsilon}$ is not necessarily a Gaussian distribution:

$$z_t = k_t * z_0 = \sqrt{\bar{\alpha}_t} z_0 + \sqrt{1 - \bar{\alpha}_t} \hat{\epsilon}. \tag{8}$$

### 3.4 Training data preparation

As discussed in 3.3, if we can group $\{z_0, z_1, z_2, ... z_t\}$ for each sample $z_t$, then the training naturally satisfies the consistency model's objective, and thus facilitates the one-step sampling. If we do not know each sample's definite backward trajectory and just randomly add the blur kernel to the clean image during training, the actual direct mapping between a blurry one and its clean version will degrade, which requires a multi-step inference for sample quality.

Our target is to build a dataset with each blurry sample grouped with its definite backward trajectory, which is not difficult for both the blur-kernel generation method and the consecutive-frame averaging method. In this work, we utilize the widely used GoPro (Nah et al., 2017) dataset. The dataset utilizes a 240fps GOPRO camera and an average of 7 - 13 successive frames to generate the blurry image for training and 11 frames for testing, while the middle frame is regarded as the sharp image. Firstly, we construct a mapping from the number of the averaging frames (n) to the timestep of the diffusion (t), with projection function $t = g(n) = (n - 1) \times 20$, which satisfies the boundary condition $g(1) = 0$, i.e., the averaged single frame is regarded as $t = 0$, and also corresponds to $z_0$ as the initial point of the forward process. Apart from that, as shown in Tab. 1, the original GoPro dataset has an unrich data distribution with most of the data averaging with 11 frames. In order to meet the requirements for consistency training, we manually enlarge the GoPro dataset to ensure that each blurry image has at least 3 points on its backward trajectories. The new statistics are shown at the bottom of Tab. 1. The detailed synthetic process is listed in the supplementary material.

## 4 Methodology

In this section, we propose FideDiff, as shown in Fig. 4. We introduce our foundation model in Sec. 4.1, and the Kernel ControlNet with kernel estimation and timestep prediction in Sec. 4.2. At last, we clarify the whole training pipeline and the loss definition in Sec. 4.3.

### 4.1 Deblurring Foundation model

Based on Stable Diffusion (Rombach et al., 2022; Esser et al., 2024), FLUX (Labs, 2024), etc, the image restoration field has witnessed significant progress from a generative perspective. However, ensuring the rationality and effectiveness of few/one-step models (Wu et al., 2024; Li et al., 2025a), while maintaining fidelity, remains an urgent issue.

According to the analysis in Sec. 3.3, we perform consistency training to enforce that the model produces temporally consistent estimates of $z_0$ for noisy latents sampled along the same trajectory, thereby enabling single-step inference. Given a training pair $(I_{LQ}, I_{HQ}, t = g(n))$, we first utilize the Variational AutoEncoder (VAE) to encode $I_{LQ}$ with the downsampling factor $d$ and gain the latent representation $z_{LQ}$, i.e., $z_t$ in Eq. 8. Then, the latent diffusion model takes $z_t$ and $t$ as input to predict $\hat{\epsilon} = \epsilon_\theta(z_t, t, c)$, where $c$ is a learnable text embedding, omitting the tokenizer and text encoder for better efficiency. As defined in Eq. 8, we obtain the predicted $\hat{z}_0$ via $\hat{z}_0 = \frac{z_t - \sqrt{1 - \bar{\alpha}_t} \hat{\epsilon}}{\sqrt{\bar{\alpha}_t}}$ and decode it as $\hat{I}_{HQ}$. We will clarify the timestep $\hat{t}$ used in the inference phase in Sec. 4.2.

To further pursue the fidelity, we abandon the various distillation methods (Wang et al., 2024; Wu et al., 2024) that are intended for more natural content generation, and adopt a GAN discriminator $\mathcal{D}$ (Sauer et al., 2024; 2025) to ensure that the generated samples closely match the real data distribution. Given the restored $\hat{z}_0$ and $z_{HQ}$ (encoded by VAE from $I_{HQ}$), the discriminator is responsible for distinguishing between the real high-quality representation $z_{HQ}$ and the generated $\hat{z}_0$, providing feedback to the generator to refine the restoration process and enhance the fidelity of the

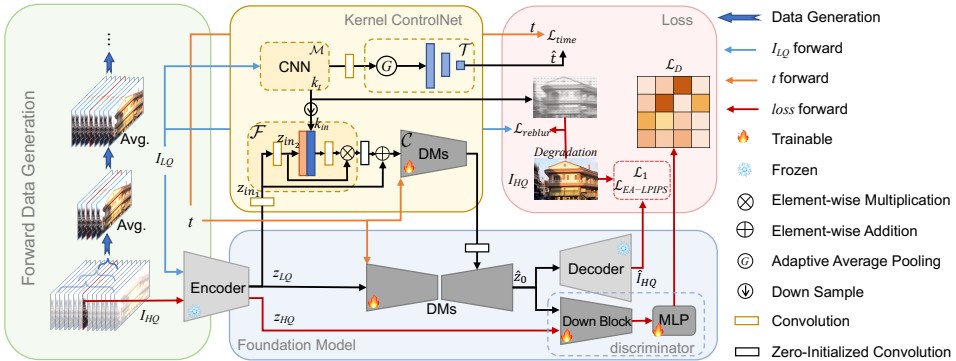

Figure 4: Training pipeline of our FideDiff. Optimized $\hat{t}$ is used for inference.

reconstructed data. The GAN discriminator consists of a pretrained UNet encoder (initialized from DMs' encoder) and several convolution blocks to generate true or false.

## 4.2 Kernel ControlNet

End-to-end learning often overlooks crucial motion information, and incorporating kernel priors into DMs remains underexplored. For instance, Liu et al. (2024b) employs two vision encoders to extract semantic information as a plugin, whereas the vanilla ControlNet (Zhang et al., 2023) accepts conditions like human pose, depth for generation, but has not explored kernel information. Recently, Lin et al. (2024) adopts a two-stage reconstruction approach and proposes IRControlNet as a post-processing modifier, enhancing the quality of the repaired image. The foundation model alone is far from sufficient for high-fidelity deblurring tasks. In Fig. 3, with estimated $k_t$, the network $\epsilon_\theta(z_t, t, c, k_t)$ is expected to be more powerful to predict the $z_0$ given the fact $z_t = k_t * z_0$.

To address this, we design Kernel ControlNet, aiming to inject kernel information as an additional condition into the Unet of the DMs. Specifically, given a blurry input $I_{LQ}$, we process the blurry image with a convolutional Unet $\mathcal{M}$ in the image space, resulting in a blur kernel representation $k_t = \mathcal{M}(I_{HQ}), k_t \in \mathcal{R}^{m \times m \times H \times W}$, where $m$ represents the predefined kernel size. Then, we do not adopt the direct mapping used in the original ControlNet where the condition is mapped to $k_{in}$ and added to $z_{in_1} = \text{Conv}(z_t)$, because the kernel map, unlike depth or pose, does not have a direct spatial correspondence with the target image. Instead, we apply a filter-like module $\mathcal{F}$:

$$z_{in_2} = \text{Conv}(z_{in_1}), \quad W = \text{Conv}(\text{Cat}(k_{in}, z_{in_2})),$$
$$O = W \otimes z_{in_2}, \quad z_{out} = z_{in_1} + Z(O),$$

where $Z$ is a convolution initialized with zeros, W is the attention weight, and $\otimes$ denotes the element-wise multiplication. Then, $z_{out}$ serves as the input for the subsequent ControlNet $\mathcal{C}$. The structure of the ControlNet remains the same as the original DM's encoder, and the initial parameters of ControlNet are copied from those of the encoder in DMs.

**t-prediction.** Additionally, Kernel ControlNet plays another crucial role in estimating the timestep $t$ during the inference phase. As discussed in Sec. 3.3, consistency training enables one-step sampling during inference. However, $t$ remains unknown at this stage. To address this, we design a small regression model $\mathcal{T}$ that follows the kernel estimation network $\mathcal{M}$ to get the $\hat{t} = \mathcal{T}(\mathcal{M}(I_{HQ}))$. Specifically, the more complex and prolonged the kernel trajectory, the higher the blur level, and consequently, the larger the timestep $t$ becomes.

## 4.3 Training Pipeline

Our training has three stages. The first stage is the foundation model training:

$$\mathcal{L} = \mathcal{L}_1(\hat{I}_{HQ}, I_{HQ}) + \lambda_1 \mathcal{L}_{EA\text{-}LPIPS}(\hat{I}_{HQ}, I_{HQ}) + \lambda_2 \mathcal{L}_G(\hat{I}_{HQ}), \tag{9}$$
$$\mathcal{L}_G = -\mathbb{E}_{t, I_{LQ}}(\log \mathcal{D}(\hat{z}_{HQ})), \tag{10}$$
$$\mathcal{L}_D = -\mathbb{E}_{t, I_{HQ}}(\log \mathcal{D}(z_{HQ})) - \mathbb{E}_{I_{LQ}}(\log(1 - \mathcal{D}(\hat{z}_{HQ}))). \tag{11}$$

The second stage is the kernel estimation network $\mathcal{M}$'s pretraining. We define the reblur loss as

$$\mathcal{L}_{reblur} = \mathcal{L}_1(\mathcal{M}(I_{HQ}) * I_{HQ}, I_{LQ}), \tag{12}$$

which uses the pixel-wise kernel estimation map to convolve the clean image, thereby regulating the kernel estimation via backpropagation. For the third stage, the foundation model is frozen, and only

| Dataset | Metrics | Restormer Zamir et al. (2022) | HI-Diff Chen et al. (2023) | UFPNet Fang et al. (2023) | AdaRevD Mao et al. (2024) | DiffBIR Lin et al. (2024) | OSEDiff Wu et al. (2024) | Diff-Plugin Liu et al. (2024b) | UID-Diff* Cheng et al. (2025) | FideDiff |
|---|---|---|---|---|---|---|---|---|---|---|
| GoPro | PSNR↑ | 32.92 | 33.33 | 34.09 | 34.60 | 26.15 | 24.34 | 22.88 | 25.08 | **28.79** |
| | SSIM↑ | 0.9611 | 0.9642 | 0.9686 | 0.9716 | 0.8377 | 0.8228 | 0.7798 | 0.7403 | **0.9148** |
| | LPIPS↓ | 0.0841 | 0.0799 | 0.0764 | 0.0712 | 0.2366 | 0.1738 | 0.2332 | 0.1310 | **0.0831** |
| | DISTS↓ | 0.0724 | 0.0710 | 0.0666 | 0.0672 | 0.1460 | 0.0834 | 0.1166 | N/A | **0.0525** |
| HIDE | PSNR↑ | 31.22 | 31.46 | 31.74 | 32.35 | 25.94 | 23.20 | 21.94 | N/A | **27.28** |
| | SSIM↑ | 0.9423 | 0.9446 | 0.9471 | 0.9525 | 0.8216 | 0.7611 | 0.7272 | N/A | **0.8775** |
| | LPIPS↓ | 0.1082 | 0.1055 | 0.0931 | 0.0872 | 0.2091 | 0.2208 | 0.2732 | N/A | **0.1068** |
| | DISTS↓ | 0.0730 | 0.0725 | 0.0676 | 0.0666 | 0.1246 | 0.1001 | 0.1411 | N/A | **0.0647** |
| RealBlur-J | PSNR↑ | 28.96 | 29.15 | 29.87 | 30.12 | 26.92 | 26.83 | 25.77 | 22.76 | **28.96** |
| | SSIM↑ | 0.8786 | 0.8898 | 0.8840 | 0.8945 | 0.7450 | 0.8004 | 0.7711 | 0.7693 | **0.8695** |
| | LPIPS↓ | 0.1561 | 0.1470 | 0.1441 | 0.1408 | 0.2587 | 0.1793 | 0.2055 | 0.1379 | **0.1142** |
| | DISTS↓ | 0.1112 | 0.1050 | 0.1101 | 0.1037 | 0.1599 | 0.1198 | 0.1355 | N/A | **0.0800** |
| RealBlur-R | PSNR↑ | 36.19 | 36.28 | 36.25 | 36.53 | 32.60 | 33.54 | 32.64 | 22.47 | **36.01** |
| | SSIM↑ | 0.9572 | 0.9583 | 0.9528 | 0.9570 | 0.8493 | 0.9056 | 0.8496 | 0.7384 | **0.9424** |
| | LPIPS↓ | 0.0608 | 0.0602 | 0.0615 | 0.0621 | 0.3388 | 0.1057 | 0.1422 | 0.2348 | **0.0584** |
| | DISTS↓ | 0.0833 | 0.0814 | 0.0854 | 0.0846 | 0.2576 | 0.1322 | 0.1673 | N/A | **0.0862** |

Table 2: Comparison results with full-reference metrics on the four datasets.

the Kernel ControlNet $\{\mathcal{F}, \mathcal{C}, \mathcal{M}, \mathcal{T}\}$ is optimized:

$$\mathcal{L} = \mathcal{L}_1(\hat{I}_{HQ}, I_{HQ}) + \lambda_1 \mathcal{L}_{EA\text{-}LPIPS}(\hat{I}_{HQ}, I_{HQ}) + \lambda_3 \mathcal{L}_{reblur} + \lambda_4 \mathcal{L}_{time}(t, \mathcal{T}(\mathcal{M}(I_{HQ}))). \quad (13)$$

EA-LPIPS is the LPIPS loss with an edge detection model, proven useful in many works (Wang et al., 2025; Li et al., 2025b). $\mathcal{L}_{time}$ is a simple $l_2$ regression loss, detailed in the supplementary.

## 5 EXPERIMENTS

### 5.1 EXPERIMENT SETTINGS

**Dataset and Evaluation.** In line with previous studies, we utilize the GoPro (Nah et al., 2017), HIDE (Shen et al., 2019), and RealBlur (Rim et al., 2020) datasets. The GoPro dataset consists of 2,103/1,111 sharp-blurry pairs for training and testing. The HIDE dataset contains 2,025 testing pairs. The RealBlur dataset has two subsets, J and R, each with 980 testing pairs. FideDiff is trained on the GoPro training set and evaluated on the four test sets as others. For evaluation, we employ the full-reference metrics: PSNR, SSIM, LPIPS (Zhang et al., 2018), and DISTS (Ding et al., 2020).

**Implementation Details.** FideDiff is based on Stable Diffusion (SD) 2.1 base version and fine-tuned with LoRA (Hu et al., 2022) in the first stage, and Kernel ControlNet is trained with full-parameter training. Because GoPro, HIDE, and RealBlur are low-resolution and affected by shooting equipment and techniques, they are unsuitable for direct SD use, especially when VAE compression loses details. To fix this, we cut the VAE downsampling from $d = 8$ to $d = 4$ by $2\times$ upsampling inputs before FideDiff and resizing outputs back. Our experiments are conducted on four NVIDIA A800-80GB GPUs. More implementation details and discussion are provided in the supplementary material. To ensure a fair comparison, we train FideDiff on the original GoPro and other models on the enlarged GoPro, as detailed in Sec. 5.3. The GoPro/HIDE test set is originally synthesized by **11** consecutive frames, so $\mathbf{t_{GT} = 200}$. RealBlur, being real-world captured, lacks a $t_{GT}$. By default, we use $\hat{t} = 200$ for GoPro/HIDE and the $t$-prediction strategy for RealBlur during inference.

### 5.2 EVALUATIONS

We compare our model, FideDiff, with many models in Tab. 2, including transformer-based models: Restormer (Zamir et al., 2022), HI-Diff (Chen et al., 2023), UFPNet (Fang et al., 2023), AdaRevD (Mao et al., 2024), and pretrained-diffusion-based DiffBIR (Lin et al., 2024), OSEDiff (Wu et al., 2024), Diff-Plugin (Liu et al., 2024b), UID-Diff (Cheng et al., 2025). DiffBIR/OSEDiff/Diff-Plugin/UID-Diff are based on the SD v2.1-base/v2.1-base/v1.4/v1.5, respectively. Diff-Plugin is originally trained on GoPro, while OSEDiff is retrained by us on GoPro. Diff-BIR is trained on a 15M dataset to address almost any type of content loss, and we combine it with AdaRevD to enhance its results. UID-Diff* uses a larger-scale unpaired deblur dataset including GoPro, and as it is closed-source, the metrics in Tab. 2 are directly copied from their paper.

**Quantitative Results.** As shown in Tab. 2, our model, FideDiff, outperforms diffusion-based methods across four full-reference metrics by a large margin. Additionally, for perceptual similarity metrics (LPIPS/DISTS), our model surpasses four transformer-based methods on at least half of the metrics across all four datasets. Notably, when evaluated on the real-world dataset RealBlur, our model demonstrates robust generalization capabilities, with perceptual similarity outperforming

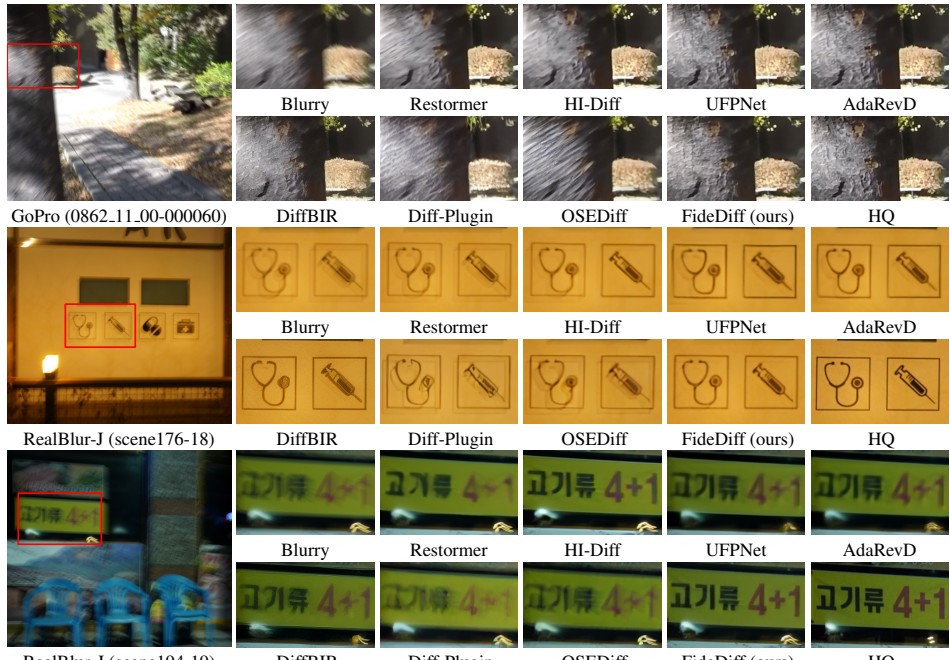

Figure 5: Visual comparison with the transformer-based and diffusion-based models.

| Model | GoPro | RealBlur-J |
|---|---|---|
| Restormer | 1.14 | 0.64 |
| HI-Diff | 1.02 | 0.58 |
| UFPNet | 0.75 | 0.42 |
| AdaRevD | 1.09 | 0.66 |
| DiffBIR-s50 | 25.40 | 12.84 |
| UID-Diff-s30* | 25 | N/A |
| Diff-Plugin-s20 | 5.29 | 2.48 |
| OSEDiff-s1 | 0.32 | 0.19 |
| Ours (d=8, w/o KCN) | 0.25 | 0.14 |
| Ours (d=4, w/o KCN) | 1.28 | 0.60 |
| Ours (d=4) | 1.52 | 0.72 |

Table 3: Inference speed (sec/image).

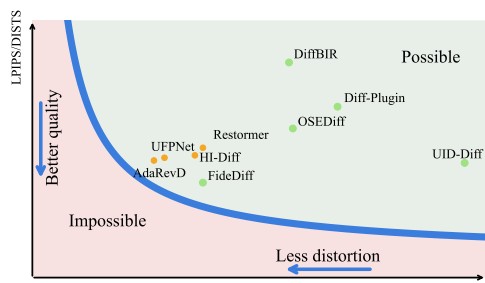

Figure 6: Perception-distortion comparisons.

most models, while the distortion metrics PSNR/SSIM show a significant reduction in the gap compared to transformer-based methods. These results indicate that our model not only better teaches the pretrained model to handle diverse types of blur but also generalizes significantly to real-world scenarios, making a crucial step towards high-fidelity image motion deblurring. Furthermore, we plot the perception-distortion tradeoff (Blau & Michaeli, 2018) curves for different models, as shown in Fig. 6. This further demonstrates that our model is closer to the original image in terms of perception, offering a balanced alternative to transformer-based methods.

**Speed.** In Tab. 3, our foundation model with downsampling factor $d = 8$ achieves the fastest inference speed. As mentioned in Sec. 5.1, influenced by the capturing devices and the low resolution of the datasets, we set $d = 4$, sacrificing time for reduced detail loss. The introduction of Kernel ControlNet further lowers the speed, but it is still comparable to most transformer-based methods and significantly faster than multi-step DMs, achieving up to a $17\times$ speedup.

**Visualizations.** Visual comparisons are shown in Figs. 1 and 5. It is evident that FideDiff significantly outperforms diffusion-based methods, with restored details closer to the ground truth compared to transformer-based methods. More visualizations are shown in the supplementary material.

## 5.3 ABLATION STUDY

**Foundation Model.** In Tab. 4, we present an ablation study of our foundation model on the original GoPro dataset, showing that LPIPS loss improves perceptual fidelity more effectively than DISTS, and that the GAN discriminator plays a crucial role—particularly when optimizing for the DISTS metric. In addition, we demonstrate that using a learnable text embedding (LE) yields better perfor-

| d | size | $l_p$ | GAN | LE | PSNR↑ | SSIM↑ | LPIPS↓ | DISTS↓ |
|---|---|---|---|---|---|---|---|---|
| 8 | 384 | LPIPS | ✓ | ✓ | 26.12 | 0.8609 | 0.1119 | 0.0648 |
| 8 | 384 | EA-LPIPS | ✓ | ✗ | 26.07 | 0.8601 | 0.1138 | 0.0641 |
| 8 | 384 | EA-LPIPS | ✗ | ✓ | 26.21 | 0.8631 | 0.1098 | 0.0732 |
| 8 | 384 | EA-DISTS | ✓ | ✓ | 25.58 | 0.8470 | 0.1288 | 0.0601 |
| 8 | 384 | EA-LPIPS | ✓ | ✓ | 26.26 | 0.8636 | 0.1093 | 0.0633 |
| 4 | 384 | EA-LPIPS | ✓ | ✓ | 27.77 | 0.8970 | 0.1020 | 0.0633 |
| 4 | 512 | EA-LPIPS | ✓ | ✓ | **28.51** | **0.9101** | **0.0899** | **0.0555** |

Table 4: Foundation model ablation.

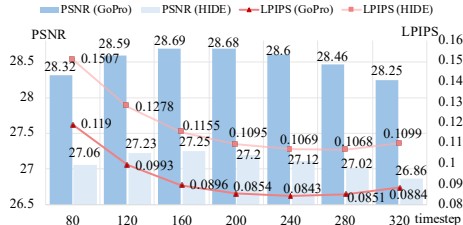

Figure 7: Performance w.r.t steps.

Table 5: Kernel ControlNet's effectiveness.

| Model | GoPro | | | | HIDE | | | |
|---|---|---|---|---|---|---|---|---|
| | PSNR↑ | SSIM↑ | LPIPS↓ | DISTS↓ | PSNR↑ | SSIM↑ | LPIPS↓ | DISTS↓ |
| base | 28.68 | 0.9130 | 0.0854 | 0.0533 | 27.20 | 0.8761 | 0.1094 | 0.0659 |
| controlnet | 28.73 | 0.9139 | 0.0844 | 0.0531 | 27.27 | 0.8774 | 0.1077 | 0.0654 |
| motion align | 28.75 | 0.9140 | 0.0851 | 0.0538 | **27.28** | **0.8777** | 0.1081 | 0.0658 |
| kernel addition | 28.70 | 0.9135 | 0.0835 | **0.0525** | 27.25 | 0.8770 | 0.1073 | 0.0648 |
| kernel | **28.79** | **0.9148** | **0.0831** | **0.0525** | **27.28** | 0.8775 | **0.1068** | **0.0647** |

mance than adopting a fixed text embedding. We also validate that incorporating an edge-enhanced perceptual loss leads to further performance gains. We demonstrate that using $d = 4$ significantly surpasses $d = 8$, mainly due to the limitations of the dataset, e.g., some scenes contain over 40 people at only 1280×720 resolution, causing substantial detail loss during SD processing (detailed in the supplementary). Increasing the training patch size further improves performance, but we ultimately use 512 due to GPU memory constraints.

**Kernel ControlNet.** Table 5 demonstrates the effectiveness of our Kernel ControlNet, which outperforms both the foundation model and a vanilla ControlNet (using only $z_{LQ}$ as input). We further show that our filter module is more effective than directly adding the kernel to $z_{LQ}$, especially in terms of PSNR and SSIM. We also experiment with a motion alignment module based on MISC-Filer (Liu et al., 2024a), but it is less effective than our Kernel ControlNet.

| Dataset | Metrics | w/o CT | w/ CT | w/ CT & TP |
|---|---|---|---|---|
| GoPro | PSNR↑ | 28.74 | **28.79** | 28.62 |
| | SSIM↑ | 0.9142 | **0.9148** | 0.9123 |
| | LPIPS↓ | 0.0871 | 0.0831 | **0.0828** |
| | DISTS↓ | 0.0548 | 0.0525 | **0.0522** |
| HIDE | PSNR↑ | 27.26 | **27.28** | **27.28** |
| | SSIM↑ | 0.8774 | 0.8775 | **0.8779** |
| | LPIPS↓ | 0.1192 | **0.1068** | 0.1155 |
| | DISTS↓ | 0.0705 | **0.0647** | 0.0694 |
| RealBlur-J | PSNR↑ | 28.95 | 28.95 | **28.96** |
| | SSIM↑ | 0.8687 | 0.8691 | **0.8695** |
| | LPIPS↓ | **0.1122** | 0.1151 | 0.1142 |
| | DISTS↓ | **0.0790** | 0.0804 | 0.0800 |
| RealBlur-R | PSNR↑ | 35.77 | 35.92 | **36.01** |
| | SSIM↑ | 0.9384 | 0.9414 | **0.9424** |
| | LPIPS↓ | 0.0662 | 0.0600 | **0.0584** |
| | DISTS↓ | 0.0961 | 0.0881 | **0.0862** |

Table 6: CT and TP ablation.

| Dataset | Metrics | OSEDiff-S | OSEDiff-L | FideDiff-S | FideDiff-L |
|---|---|---|---|---|---|
| GoPro | PSNR↑ | **24.34** | 24.29 | 28.56 | **28.79** |
| | SSIM↑ | 0.8228 | **0.8237** | 0.9109 | **0.9148** |
| | LPIPS↓ | 0.1738 | **0.1634** | 0.0873 | **0.0831** |
| | DISTS↓ | 0.0834 | **0.0772** | 0.0549 | **0.0525** |
| HIDE | PSNR↑ | 23.20 | **23.52** | 27.07 | **27.28** |
| | SSIM↑ | 0.7611 | **0.7779** | 0.8737 | **0.8775** |
| | LPIPS↓ | 0.2208 | **0.2137** | 0.1141 | **0.1068** |
| | DISTS↓ | **0.1001** | 0.1036 | 0.0690 | **0.0647** |
| RealBlur-J | PSNR↑ | 26.83 | **26.92** | 28.88 | **28.96** |
| | SSIM↑ | 0.8004 | **0.8044** | 0.8680 | **0.8695** |
| | LPIPS↓ | 0.1793 | **0.1758** | 0.1145 | **0.1142** |
| | DISTS↓ | 0.1198 | **0.1183** | 0.0804 | **0.0800** |
| RealBlur-R | PSNR↑ | **33.54** | 33.04 | 35.91 | **36.01** |
| | SSIM↑ | **0.9056** | 0.8937 | 0.9400 | **0.9424** |
| | LPIPS↓ | **0.1057** | 0.1074 | 0.0606 | **0.0584** |
| | DISTS↓ | **0.1322** | 0.1408 | 0.0894 | **0.0862** |

Table 7: Fair comparison.

**Consistency training (CT) and time prediction (TP).** In Tab. 6, "w/ CT" refers to our time-consistency training, where different blur levels correspond to varying $t$ values ranging from 80 to 280 (calculated with $g(n)$ according to Tab. 1). "w/o CT" means all training images are set to the same $t = 200$. When inference is set with $\hat{t} = t_{GT} = 200$, FideDiff w/ CT outperforms FideDiff w/o CT on GoPro/HIDE/RealBlur-R, especially in perceptual similarity metrics like LPIPS and DISTS. This demonstrates the effectiveness of our consistency training in better decoupling different levels of blurriness and controlling blurry details. After adding t-prediction to the CT

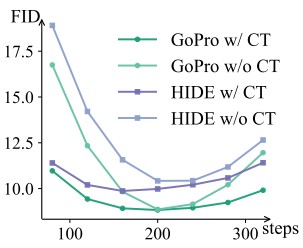

Figure 8: FID w.r.t. steps.

model, performance all improves on RealBlur, proving that t-prediction is effective for real-world blur with good generalization, further enhancing fidelity. On GoPro/HIDE, although t-prediction cannot perfectly predict $\hat{t} = t_{GT}$, its performance is comparable to using no t-prediction.

Additionally, we analyze the variation in distortion (PSNR, SSIM) and perception (LPIPS, DISTS) of our foundation model w/ CT at different timesteps on GoPro/HIDE in Fig. 7. Despite $t_{GT} = 200$, when $\hat{t}$ varies between approximately 120 and 280, the model maintains good performance, with either perception or distortion being better than at $\hat{t} = 200$. This demonstrates the robustness and desirable properties of our model, which are not presented w/o CT (in the supplementary material). Furthermore, we plot the FID curves of two models on the GoPro/HIDE datasets in Fig. 8, where w/CT demonstrates superior distribution similarity across all timesteps and maintains good generalization over a wider range. The effectiveness of CT and TP inspires us to see how DMs can deal more effectively with varying degradation levels. The correct identification of degradation levels is crucial for achieving accurate model performance, as treating all levels uniformly is not effective.

**Fair Comparison.** We train FideDiff on the original GoPro (S) and OSEDiff on our enlarged GoPro (L). As shown in Tab. 7, FideDiff-S still outperforms the diffusion baseline and can effectively leverage the abundant data to further boost performance. We also show the AdaRevD retraining results in the supplementary material, where it also struggles to fully learn the enlarged data.

## 6    CONCLUSION

In this paper, we analyze the challenges faced by current DMs in real-world deployment, focusing on time efficiency and fidelity. We reformulate the forward and backward processes of deblurring, design a time-consistent training paradigm, and develop FideDiff for high-fidelity single-step deblurring. We also introduce Kernel ControlNet to inject blur kernel conditions into the foundation model for enhanced fidelity, and we predict the timestep based on kernel estimation to dynamically select it during inference. Our model demonstrates strong performance in evaluations, advancing rapid, high-fidelity restoration of DMs in real-world applications and driving the field's progress.

## ACKNOWLEDGEMENTS

This work is supported by the National Natural Science Foundation of China (62501386, 625B2116) and CCF-Tencent Rhino-Bird Open Research Fund.

## ETHIC STATEMENT

All authors confirm adherence to the ICLR Code of Ethics throughout submission and discussion.

## REPRODUCIBILITY STATEMENT

We provide a detailed description of the training and testing procedure in the supplementary material and will release the code and dataset.

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
