# Supplementary Material:
# FideDiff: Efficient Diffusion Model for High-Fidelity Image Motion Deblurring

**Xiaoyang Liu**[1][*] **Zhengyan Zhou**[1][*] **Zihang Xu**[1],
**Jiezhang Cao**[2], **Zheng Chen**[1], **Yulun Zhang**[1][†]
[1]Shanghai Jiao Tong University, [2]Harvard University

## A    More implementation details

Our model is trained in three stages, with 100k, 50k, and 50k steps for each stage. In the first stage, we fine-tune using LoRA with a LoRA rank of 96, employing a progressive learning strategy (Zamir et al., 2022). The patch sizes are [160, 192, 256, 320, 384, 448, 512], and the corresponding batch sizes are [32, 32, 16, 16, 16, 8, 8]. We train for [8k, 8k, 8k, 8k, 8k, 10k, 10k] steps, and the remaining 40k steps are trained with a patch size of 512 and batch size 8. The loss coefficients are set as $\lambda_1 = 1$ and $\lambda_2 = 5 \times 10^{-3}$ to ensure that their contributions to the total loss are on the same scale. In the second stage, we train for 50k steps with a batch size of 48. In the third stage, we merge the LoRA parameters from the first stage into the original SD 2.1 base UNet and initialize the ControlNet with these parameters. The kernel estimation module is also initialized using the second-stage results, with $\lambda_3 = 0.1$, and we train for another 50k steps.

For the time-prediction module, instead of directly regressing the timestep, we apply a sigmoid function to the output of the final layer, mapping it to the range [0, 1]. As described in the main text, the GoPro synthesis includes averaging frames [5, 7, 9, 11, 13, 15], with corresponding timesteps calculated using $g(n) = (n - 1) \times 20$, resulting in [80, 120, 160, 200, 240, 280]. We define a mapping dictionary that associates these timesteps with values in [0.15, 0.29, 0.43, 0.57, 0.71, 0.85], maintaining the same interval. We compute the loss by comparing the sigmoid output of the model with the mapped $t_{GT}$ values, using L2 loss with a scaling factor of $\lambda_4 = 1$. During the inference stage, after obtaining predictions normalized between [0,1], they are linearly scaled to the true timesteps $\hat{t}$ and then rounded to integer values. For the RealBlur-R dataset, due to its excessively dark scenes, we introduced a fixed offset term to the time prediction for correction.

**VAE downsampling factor.**    Here, we explain why we set the VAE downsampling factor $d$ to 4 during both training and inference, instead of the default value of 8. As is well known, the advent of latent diffusion models is aimed at significantly compressing images into latent space to accelerate the inference process. However, the issue lies in the fact that VAE compression is not lossless. VAE training is typically fine-tuned on images of a fixed size (e.g., 512) with less complex scenes. In contrast, datasets in the deblurring domain, such as GoPro, HIDE, and RealBlur, are constrained by the limitations of the capturing devices, with complex scenes but very low resolution, typically 1280×720 or 680×760. Such images are not suitable for direct processing by latent diffusion models. For example, in some GoPro and HIDE datasets, faces occupy only 10–20 pixels, and after VAE compression, they become a single point in the latent space, making it impossible to recover the original details. We aim to improve reconstruction accuracy at a low cost. Thus, we upsample the deblur dataset by a factor of two before passing it through FideDiff, and then resize the resulting image back to its original dimensions. This strategy is equivalent to altering the downsampling factor of the VAE. This also inspires us to design a dynamic VAE that adapts to different levels of image complexity by setting different downsampling factors $d$. We leave this for future research.

We demonstrate the importance of upsampling in Tab. 1. The data shown in the table separately compares the similarity of low-quality and high-quality images reconstructed by directly compressing and rebuilding using the VAE of the Stable Diffusion 2.1 base, relative to the original input. As observed, the VAE reconstruction of high-quality images achieves only 30.44 dB, which is 4 dB lower than the current SOTA AdaRevD (Mao et al., 2024). Upsampling the input before reconstruction alleviates this issue to some extent.

---

[*]Equal contribution
[†]Corresponding author: Yulun Zhang, yulun100@gmail.com

| Upsampling Factor | ×1 | ×2 | ×3 |
|---|---|---|---|
| Low-quality | 36.91 | 41.72 | 42.71 |
| High-quality | 30.44 | 36.75 | 39.19 |

Table 1: PSNR in VAE Reconstruction of GoPro Images.

## B  MORE ABLATIONS

We present all the choices for our foundation model in Tab. 2. In the original setting, which is used in works such as (Wang et al., 2025; Li et al., 2025), L2 and EA-DISTS are employed, but they demonstrate limited fidelity to high-quality images. The notation $4^*$ indicates that we train on a 384 patch size with a downsampling factor $d = 8$ and directly apply $d = 4$ during inference. As shown, direct upsampling without upsampling during training can improve PSNR and SSIM, but it results in worse performance in LPIPS and DISTS. Only when we train with $d = 4$ and perform inference with the same factor can we fully enhance performance. Additionally, we clarify that the size listed in Tab. 2 refers to the final SD input patch size, meaning that a 512 patch with $d = 4$ in the table corresponds to a 256 patch in the original image.

| $l_{pixel}$ | $l_{percep}$ | d | size | GAN | GoPro PSNR↑ | SSIM↑ | LPIPS↓ | DISTS↓ | RealBlur-J PSNR↑ | SSIM↑ | LPIPS↓ | DISTS↓ |
|---|---|---|---|---|---|---|---|---|---|---|---|---|
| $l_2$ | DISTS | 8 | 384 | ✓ | 25.18 | 0.8379 | 0.1311 | 0.0599 | 27.25 | 0.8040 | 0.1412 | 0.0941 |
| $l_1$ | DISTS | 8 | 384 | ✓ | 25.58 | 0.8470 | 0.1288 | 0.0601 | 27.37 | 0.8092 | 0.1400 | 0.0946 |
| $l_2$ | LPIPS | 8 | 384 | ✓ | 26.08 | 0.8606 | 0.1094 | 0.0627 | 27.68 | 0.8188 | 0.1257 | 0.0907 |
| $l_1$ | LPIPS | 8 | 384 | ✓ | 26.26 | 0.8636 | 0.1093 | 0.0633 | 27.73 | 0.8203 | 0.1241 | 0.0896 |
| $l_1$ | LPIPS | 8 | 384 | ✗ | 26.21 | 0.8631 | 0.1098 | 0.0732 | 27.68 | 0.8190 | 0.1243 | 0.0950 |
| $l_1$ | LPIPS | 4* | 384 | ✓ | 27.26 | 0.8876 | 0.1143 | 0.0700 | 28.17 | 0.8358 | 0.1306 | 0.0942 |
| $l_1$ | LPIPS | 4 | 384 | ✓ | 27.77 | 0.8970 | 0.1020 | 0.0633 | 28.49 | 0.8427 | 0.1226 | 0.0885 |
| $l_1$ | LPIPS | 4 | 512 | ✓ | **28.51** | **0.9101** | **0.0899** | **0.0555** | **28.85** | **0.8646** | **0.1166** | **0.0814** |

Table 2: Ablation of the foundation model.

We also train the AdaRevD model on our enlarged GoPro dataset. As shown in Tab. 3, AdaRevD fails to learn additional knowledge and loses some generalization capability on other datasets. Additionally, we compare the performance of our foundation model with and without CT in Tab. 4, with the timestep set to 200. As observed, with CT, our model outperforms the one without CT on GoPro, HIDE, and RealBlur-R datasets.

| Dataset | Metrics | AdaRevD-S | AdaRevD-L |
|---|---|---|---|
| GoPro | PSNR↑ | **34.60** | 33.73 |
| | SSIM↑ | **0.9716** | 0.9577 |
| | LPIPS↓ | 0.0712 | **0.0685** |
| | DISTS↓ | 0.0672 | **0.0588** |
| HIDE | PSNR↑ | **32.35** | 31.43 |
| | SSIM↑ | **0.9525** | 0.9263 |
| | LPIPS↓ | **0.0872** | 0.1192 |
| | DISTS↓ | **0.0666** | 0.0687 |
| RealBlur-J | PSNR↑ | **30.12** | 29.73 |
| | SSIM↑ | **0.8945** | 0.8687 |
| | LPIPS↓ | **0.1408** | 0.1510 |
| | DISTS↓ | **0.1037** | 0.1116 |
| RealBlur-R | PSNR↑ | **36.53** | 35.93 |
| | SSIM↑ | **0.9570** | 0.9442 |
| | LPIPS↓ | **0.0621** | 0.0952 |
| | DISTS↓ | **0.0846** | 0.1476 |

Table 3: AdaRevD results trained on our enlarged GoPro dataset.

| Dataset | Metrics | Base w/o CT | Base w/ CT |
|---|---|---|---|
| GoPro | PSNR↑ | 28.65 | **28.68** |
| | SSIM↑ | **0.9130** | **0.9130** |
| | LPIPS↓ | 0.0892 | **0.0854** |
| | DISTS↓ | 0.0545 | **0.0533** |
| HIDE | PSNR↑ | 27.17 | **27.20** |
| | SSIM↑ | 0.8760 | **0.8761** |
| | LPIPS↓ | 0.1240 | **0.1094** |
| | DISTS↓ | 0.0716 | **0.0659** |
| RealBlur-J | PSNR↑ | **28.94** | 28.87 |
| | SSIM↑ | **0.8660** | 0.8652 |
| | LPIPS↓ | **0.1142** | 0.1180 |
| | DISTS↓ | **0.0800** | 0.0819 |
| RealBlur-R | PSNR↑ | 35.60 | **35.86** |
| | SSIM↑ | 0.9336 | **0.9391** |
| | LPIPS↓ | 0.0705 | **0.0583** |
| | DISTS↓ | 0.0980 | **0.0855** |

Table 4: Results of the foundation model w/ and w/o consistency training (CT).

In Fig. 1 and Fig. 2, we present the performance of the foundation model of FideDiff with and without consistency training (CT) at different time steps. As shown, without CT, the model performs well only at the training step $t = 200$, but as $t$ deviates from 200, the performance rapidly deteriorates. Moreover, the optimal performance without CT does not match that of the model with CT.

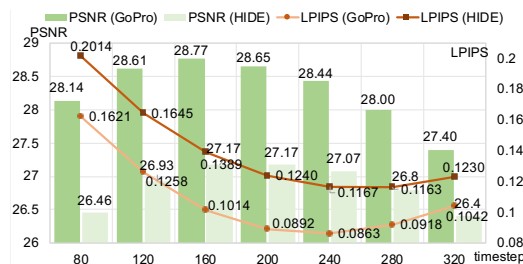

Figure 1: Foundation model's performance w/ CT w.r.t. steps on GoPro and HIDE.

Figure 2: Foundation model's performance w/o CT w.r.t. steps on GoPro and HIDE.

## C  THE STRUCTURE CLARIFICATION

In our proposed Kernel ControlNet, we employ a convolutional Unet to predict the blur kernel, following (Kim et al., 2021; Fang et al., 2022), with two encoding and decoding blocks. The encoding block includes a $3 \times 3$ convolutional layer with a stride of 2 for downsampling, while the decoding block utilizes a deconvolution layer for upsampling. The feature maps from the encoding blocks are concatenated with those from the decoding blocks via skip connections. The central convolutional module consists of 5 convolution layers, each followed by a ReLU activation function.

The kernel estimation output, denoted as $k_t \in \mathcal{R}^{m \times m \times H \times W}$, is a kernel map where $m$ is set to 19, meaning the map has dimensions of $19 \times 19$. For the t-prediction module, we first resize $k_t$ to $\mathcal{R}^{m^2 \times H \times W}$, then pass it through a linear layer followed by AdaptiveAvgPool2d to resize it to a fixed $8 \times 8$ size. After that, all dimensions except the batch dimension are flattened, followed by a sequence of a linear layer, ReLU activation, another linear layer, and a sigmoid function to normalize the output to the range [0,1].

In the main paper, we also explore the use of MISC Filter (Liu et al., 2024) in the Kernel ControlNet section. Specifically, we utilized its Motion-Guided Alignment (MGA) to obtain motion flow and masks, and leveraged this to align the content of the skip connections in the UNet of SD. However, the resulting performance is not as effective as the kernel estimation approach.

## D  DATA GENERATION

In the construction of our GoPro dataset, we generate blurry images by averaging frames based on the original GoPro samples. For the 7-frame average sample, using the middle sharp frame as a reference, we further synthesize blurry images by averaging 9 and 11 frames. For the 13-frame average sample, using the middle sharp frame as a reference, we synthesize blurry images by averaging 15 and 11 frames. For the 11-frame average sample, we employ a probabilistic augmentation method. First, we synthesize all 13-frame combinations. Then, we generate blurry images by averaging 9 and 15 frames based on a probability distribution of [0.3, 0.7]. In the second augmentation step, we further synthesize blurry images by averaging 9, 7, and 5 frames based on a probability distribution of [0.1, 0.45, 0.45] to ensure the dataset's balance.

## E  VISUALIZATIONS

We present additional visual comparison results. On the RealBlur-J dataset (Fig. 3), our deblurring method outperforms all existing approaches, including both diffusion-based and transformer-based methods, and in many cases, producing results that are closer to high-quality images. On the GoPro (Fig. 4) and HIDE (Fig. 5) datasets, our method shows a clear advantage over diffusion-based methods, such as better recovery of blurry contents and higher fidelity to the ground truth image, and achieves visual performance comparable to, or even surpassing, transformer-based methods. It is worth noting that transformer-based models sometimes lose details and may leave small amounts of blur, while DiffBIR tends to over-generate, producing details that should not exist. OSEDiff and Diff-Plugin, on the other hand, struggle to deblur and recover details in many complex scenarios. This demonstrates that our model is leading within the diffusion-based methods and is on par with well-developed transformer-based approaches. This significantly expands the application of diffusion models in low-level vision tasks, particularly image restoration, providing new insights and a strong baseline for industrial deployment and real-world deblurring.

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

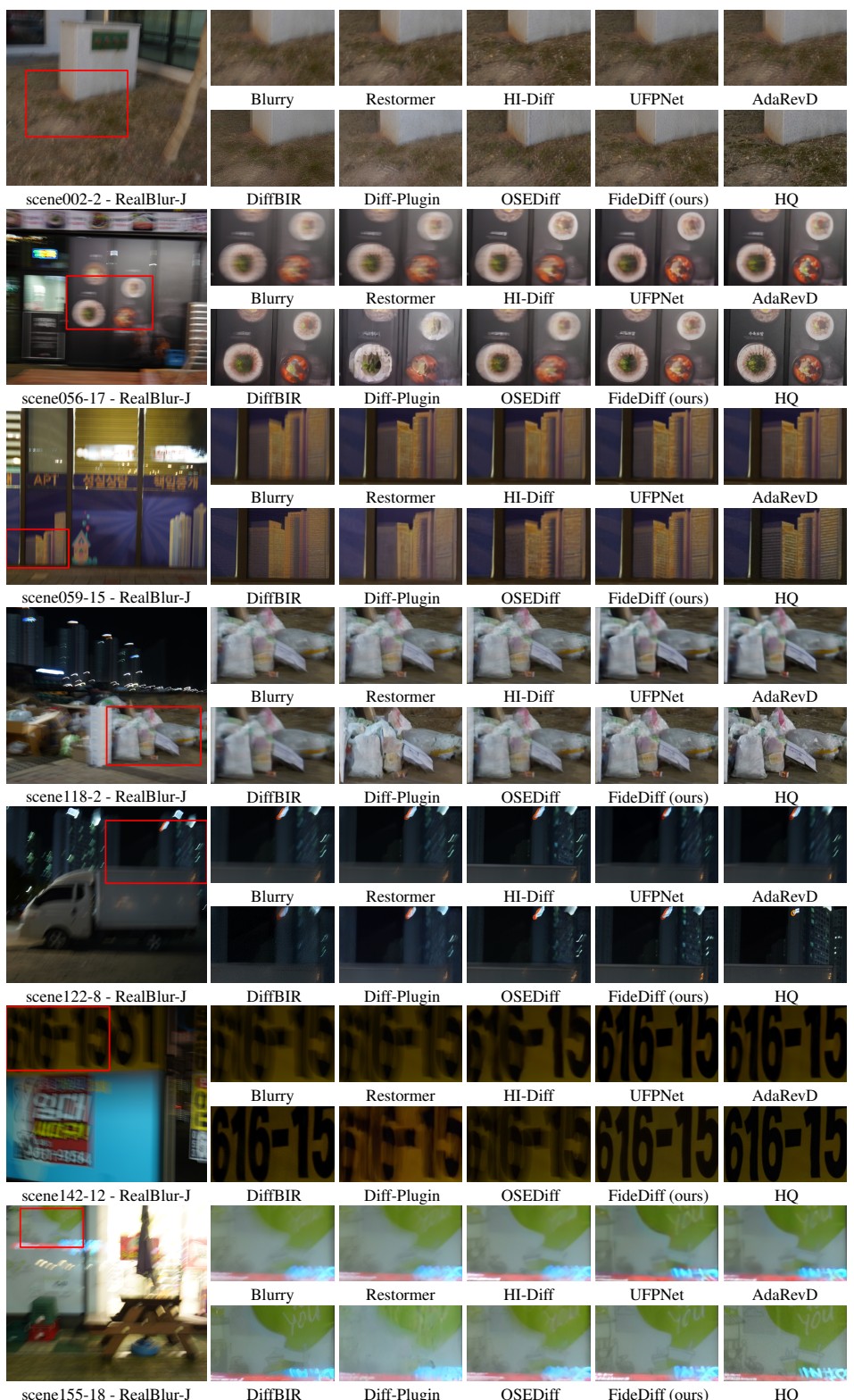

Figure 3: More visualizations on RealBlur-J.

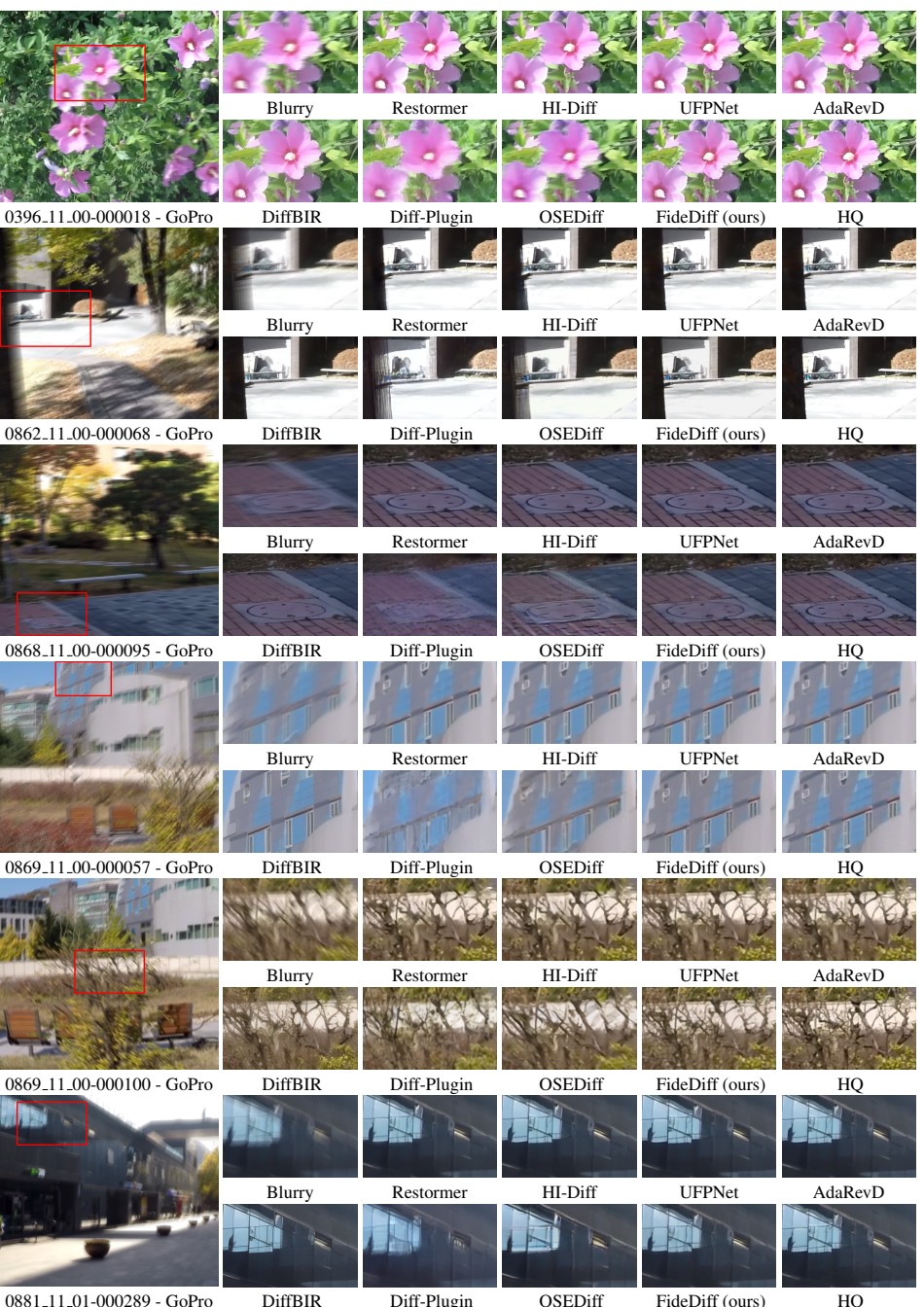

Figure 4: More visualizations on GoPro.

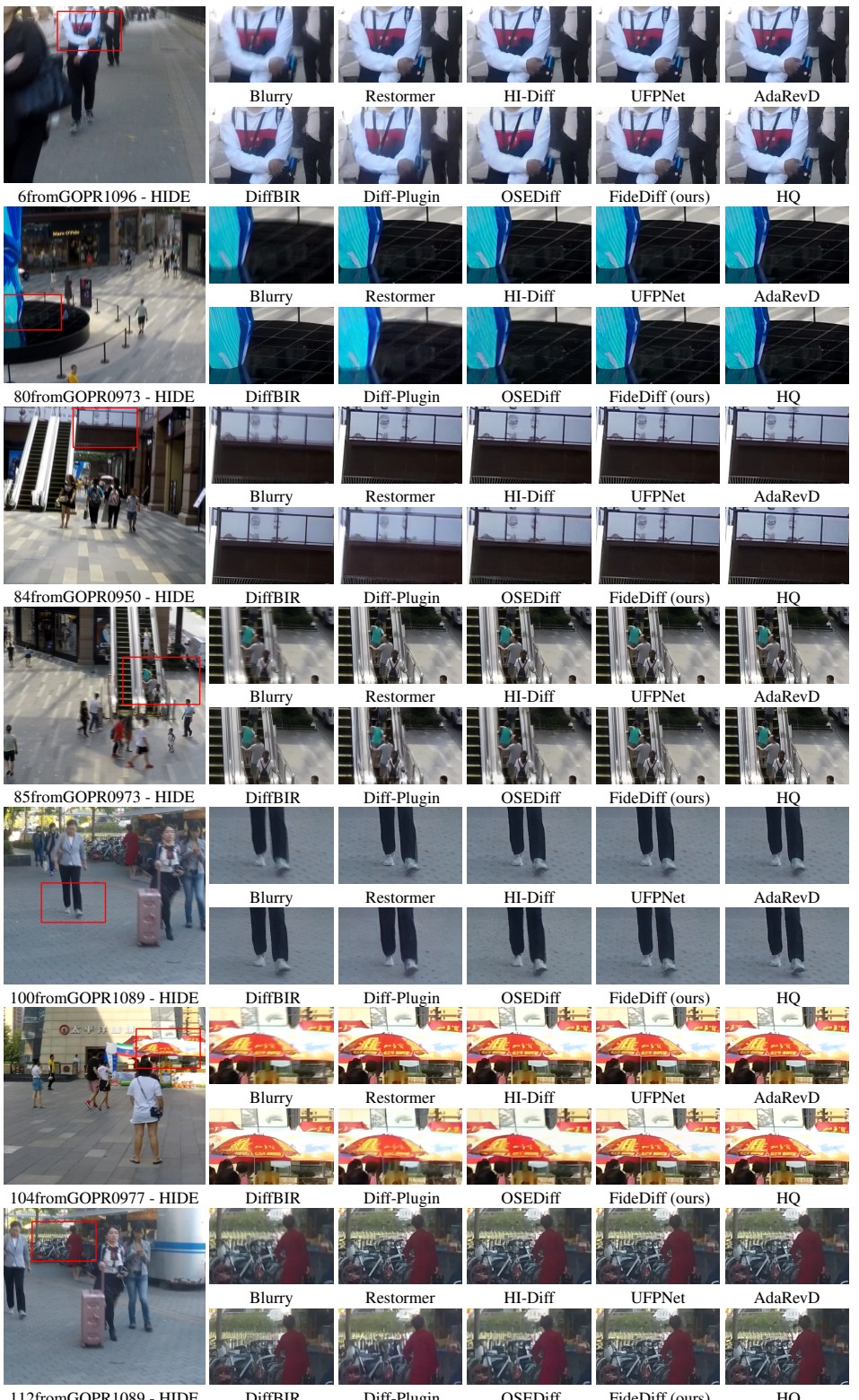

Figure 5: More visualizations on HIDE.