# OpenReview forum: "FideDiff: Efficient Diffusion Model for High-Fidelity Image Motion Deblurring"
_ICLR.cc/2026/Conference — ICLR 2026 Poster_

### Official Review · Reviewer_n57E · 2025-10-27

**Soundness:** 4
**Presentation:** 3
**Contribution:** 4
**Rating:** 6
**Confidence:** 4

**Summary:**

This paper introduces a single-step, diffusion-based deblurring model that improves the pixel and perceptual fidelity of deblurring results. Motion deblurring can be formulated as a diffusion-like process, with each timestep representing an increasingly blurred image. A kernel control net is proposed for kernel estimation and adaptive timestep prediction. This approach outperforms existing diffusion-based methods and achieves superior performance on reference-based metrics.

**Strengths:**

The proposed method is novel, and the manuscript is well organised and written. FideDiff significantly improves the fidelity of deblurring results, outperforming other diffusion-based models. It may also encourage other researchers to explore diffusion-based, high-fidelity methods for other restoration tasks.

**Weaknesses:**

1.	In Section 3.4, the mapping function between the number of averaging frames and the timestep is t = g(n) = (n - 1) * 20. Why was this set?
2.	The symbol * is used many times in the manuscript, but it has different meanings in each instance. For example, in Eq. (4), it represents a convolution operation. However, in the mapping function t = g(n), it may represent a multiplication operation. Please clarify this.
3.	The proposed framework uses a learnable text embedding as a condition for prediction. However, this is lacking in the framework architecture shown in Fig. 4. Moreover, what are the text descriptions used in the proposed method? Is the CLIP model used in the proposed framework?
4.	Figure 4 lacks the symbol k_t, which represents a blur kernel. Furthermore, Z_(in1) = Conv(Z_t) should be Z_(in1) = Conv(Z_(LQ)(Z_t)). Where is k_(in) in Fig. 4? What do the orange and blue blocks represent? In the filter-like module F, these lines lack arrows. Does * represents the element-wise multiplication? Why are there two lines with arrows showing the output of the DMs?
5.	What is the EA-LPIPS function? How does it influence the performance of image deblurring? Please conduct experiments to evaluate this.
6.	What is the meaning of * in Eq.(12)?
7.	I think the proposed method could be extended to other low-level tasks, such as super-resolution and deraining. However, it is unclear how it can be extended to image compression, given that this task does not involve modelling physical degradation.
8.	In the implementation details, I see that you perform 2x upsampling before inference and then resize the outputs back to their original spatial size. What interpolation operation is used here, and might it cause detail loss?

**Questions:**

Please See the weaknesses part.

---

> ### Author Response · Authors · 2025-11-21
> **Response to Reviewer n57E (denoted as R4) part1**
>
> **R4-Q1:** question about $t = g(n) = (n - 1)\times20$
>
> In Stable Diffusion, T = 1000 with t=0 for clean images and t=1000 for pure noise, so directly using $t \in [1,15]$ in $[0,1000]$ would place all blur levels in an uninformative tiny timestep range. Following prior diffusion works (e.g., InvSR, FluxSR, HAODiff), we anchor the common 11-frame average at t = 200 and use the simple linear mapping $t = g(n) = (n - 1) \times 20$. This yields well-separated, reasonable timesteps for different blur levels, and the factor 20 is an empirical normalization rather than a finely tuned hyperparameter. We also conduct a timestep experiment, setting the projection factor to 10 and 40, the results are as follows:
> |Factor|PSNR↑|SSIM↑|LPIPS↓|DISTS↓|
> |------|-----|-----|------|------|
> | 10 | 28.42 | 0.9088 | 0.0912 | 0.0568 |
> | **20**  | 28.51 | 0.9101 | 0.0899 | 0.0555 |
> | 40 | 28.48 | 0.9097 | 0.0907 | 0.0559 |
>
> **R4-Q2:** symbol *
>
> Thank you for pointing this out! We have revised the main text: in Eq. 4, the operator * denotes a convolution operation, and in the mapping function we now use the multiplication symbol “$\times$”.
>
> **R4-Q3:** learnable text embedding
>
> Our text embedding is a **learnable conditioning vector** that is optimized jointly with the Foundation Model during training. For efficiency and flexibility, we **do not** use a tokenizer or a CLIP text encoder in our main framework; instead, we directly inject a learnable text embedding into the UNet as an additional condition. This design reduces computational overhead while allowing the model to learn task-specific semantic priors directly in the latent space.
>
> In addition, we conduct an ablation where we replace the learnable embedding with a fixed text description: *“a photo restored from motion blur, with sharp focus, detailed, high clarity, realistic texture, natural colors and lighting”*
>
> The results below (also added to Table 4) show that the learnable text embedding consistently improves performance over this fixed textual prompt. We will also update Fig. 4 in the revised manuscript to explicitly indicate the text-conditioning path.
>
> |Learnable Embedding|PSNR↑|SSIM↑|LPIPS↓|DISTS↓|
> |------|-----|-----|------|------|
> | x | 26.07 | 0.8601 | 0.1138 | 0.0641 |
> | ✓  | 26.26 | 0.8636 | 0.1093 | 0.0633 |
>
>
> **R4-Q4:** Question about figure 4.
>
> Thank you for the suggestion! In the revised version of the paper, we have explicitly added both $k_t$ and $k_{in}$. The blue block at the bottom left of Kernel ControlNet now corresponds to $k_{in}$, and the orange block to $z_{in_2}$. We have also added directional arrows inside module F. In module F, we replace “*” with “$\otimes$” to indicate element-wise multiplication.
>
> Regarding the two arrows from the DMs output: our original intention was to convey that ControlNet does not only inject its final output into the UNet. In fact, the features from each encoder layer of ControlNet are preserved and added to the corresponding UNet skip connections, which is the standard behavior of ControlNet. Now, to avoid ambiguity, we keep only a single arrow in the figure.
>
> As for the remark that $z_{in_1} = \text{Conv}(z_t)$ should be $z_{in_1} = \text{Conv}(z_{LQ}(z_t))$, we clarify that $z_{LQ}(z_t)$ is not intended to denote a function or separate module. Our intention is that $z_{LQ}$ and $z_t$ refer to the **same latent variable**: $z_{LQ}$ is labeled from the perspective of the restoration task (low-quality latent), while $z_t$ is labeled from the diffusion process (latent at timestep $t$). We have revised Section 4.1 to make this correspondence explicit.

---

> ### Author Response · Authors · 2025-11-21
> **Response to Reviewer n57E (denoted as R4) part2**
>
> **R4-Q5:** EA-LPIPS function.
>
> $\mathcal{L} _{EA-LPIPS} \(\hat{I} _{HQ}, I _{HQ} \) = \mathcal{L} _{LPIPS} \(\hat{I} _{HQ}, I _{HQ} \) + \mathcal{L} _{LPIPS}\(S\(\hat{I} _{HQ}\), S\(I _{HQ}\)\)$.
> The Sobel operator $S$ consists of two convolution kernels, $G_x$ and $G_y$, which detect horizontal and vertical edges, respectively:
> $
> G_x =
> \begin{bmatrix}
> -1 & 0 & 1 \\\\
> -2 & 0 & 2 \\\\
> -1 & 0 & 1
> \end{bmatrix},
> \quad
> G_y =
> \begin{bmatrix}
> -1 & -2 & -1 \\\\
> 0  &  0 &  0 \\\\
> 1  &  2 &  1
> \end{bmatrix}.
> $
> The Sobel operator is applied to an image $x$ as follows:
> $ S(x) = \sqrt{(G_x * x)^2 + (G_y * x)^2} $,
> where * denotes the convolution operation. The following experimental results demonstrate the effectiveness of the edge detection model, and we have included them in Table 4 of the main paper.
> | $l_p$ |PSNR↑|SSIM↑|LPIPS↓|DISTS↓|
> |------|-----|-----|------|------|
> | LPIPS | 26.12 | 0.8609 | 0.1119 | 0.0648 |
> | EA-LPIPS  | 26.26 | 0.8636 | 0.1093 | 0.0633 |
>
> **R4-Q6:** meaning of * in Eq.(12)?
>
> This is a convolution operation, where the pixel-wise kernel estimation map is convolved with the sharp image to compute the loss. We have added a corresponding clarification in the revised main text.
>
> **R4-Q7:**  extension to other tasks?
>
> We appreciate the reviewer’s insightful comment. Our main claim is that the proposed framework is naturally extendable to low-level restoration tasks where one can define a monotonic degradation level and a corresponding forward operator (e.g., blur strength, downsampling factor, rain density). Although image compression is not a *physical* degradation, it can still be modeled as an operator $x_Q = \mathcal{C}_Q(x_0)$ parameterized by the compression quality factor or bitrate Q. In this case, the timestep t can be defined as a monotonic function of Q, and our time-consistency training can be applied by constructing pairs of compressed images at different qualities that share the same uncompressed target. Similarly, the kernel-aware ControlNet can be generalized to a “compression-aware” ControlNet that conditions on codec-specific side information (e.g., quantization tables or spatial quality maps), in line with recent one-step diffusion models for JPEG artifact removal. This is also an important direction for our future work, where we aim to further advance high-fidelity and efficient diffusion models for a broader range of low-level vision tasks.
>
>
>
> **R4-Q8:** upsampling interpolation operation.
>
> We use the LANCZOS interpolation algorithm, which provides higher accuracy. We compare LANCZOS with BICUBIC and BILINEAR, and find that LANCZOS and BICUBIC achieve similar performance with identical FID scores, whereas BILINEAR introduces substantially larger perceptual errors (LPIPS/DISTS/FID). We will include this detail in the supplementary material.
>
> |Interpolation|PSNR↑|SSIM↑|LPIPS↓|DISTS↓|FID↓|
> |-------------|----|----|-----|-----|---|
> |  LANCZOS    | 28.79 | 0.9148 | **0.0831** | **0.0525** | **8.62** |
> |  BICUBIC    | 28.84 | 0.9151 | 0.0888 | 0.0556 | **8.62** |
> |  BILINEAR   | **28.86** | **0.9153** | 0.1191 | 0.0719 | 9.18 |

---

> > ### Comment · Reviewer_n57E · 2025-11-26
> >
> > Thank you for your detailed response. I have improved my score and hope that this paper will be accepted. Good luck to you!

---

> > > ### Author Response · Authors · 2025-11-27
> > >
> > > Dear Reviewer n57E,
> > >
> > > We sincerely appreciate your support and the improved score. We are deeply grateful for your positive feedback and kind wishes.
> > >
> > > Thank you again for your time and effort in reviewing our paper!
> > >
> > > Best regards,
> > >
> > > Authors

---

### Official Review · Reviewer_Hp3A · 2025-10-28

**Soundness:** 4
**Presentation:** 3
**Contribution:** 3
**Rating:** 8
**Confidence:** 4

**Summary:**

This paper proposed a novel motion deblurring architecture based on a pre-trained stable diffusion model, successfully distil the multi-step diffusion model into a single-step model.

**Strengths:**

1. The performance shown in the paper is promising. Even though the model is trained on synthetic data, it shows better performance on all datasets compared with other diffusion-based methods.
2. Solid mathematical analysis. Although the markov prediction of kt from kt-1 is infeasible to calculate, following the training assumption of diffusion model, every training step share the same sharp ground truth, which makes the single-step possible.
3. The paper proposed an efficient single-step diffusion model that avoid the loss of diffusion inductive bias as other one-step methods did, which reduce the sensitivity of the method to spatial-variant and severe blurs. The key thing here is the proposed time-consistency training.
4. High-fidelity results, which surpass both conventional and diffusion based methods.
5. Properly modify the controlnet to better inject the blur kernel as an extra condition.

**Weaknesses:**

1. How to generate z_0 with input Z_LQ is unclear. Needs further clarification.
2. The training data compatible of this architecture are limited, which reduces the generalization of the proposed method.
3. How to concatenate latent feature z with the estimated blur kernel?

**Questions:**

1. What is the blue block in Figure 4?
2. Why can't the model be trained end-to-end instead of using three stages.
3. Any evaluation of time prediction? I am not sure how accurate it is.
4. The data generation process in supp is unclear to me. What is the meaning of "For the 13-frame average sample, using the middle sharp frame as a reference, we synthesize blurry images by averaging 15 and 11 frames" and "For the 11-frame average sample, we employ a probabilistic augmentation method. First, we synthesize all 13-frame combinations. Then, we generate blurry images by averaging 9 and 15 frames based on a probability distribution of [0.3, 0.7]."

---

> ### Author Response · Authors · 2025-11-21
> **Response to Reviewer Hp3A (denoted as R3) part1**
>
> **R3-Q1:** How to generate z_0 with input Z_LQ?
>
> According to Equation 3 and 8, we use $\hat{z}_0 = \frac{z_t - \sqrt{1 - \bar{\alpha}_t} \hat{\epsilon}}{\sqrt{\bar{\alpha}_t}}$ to generate the $\hat{z}_0$, $ z _t $ is equal to $ z _{LQ} $. We have added this to Section 4.1 in the main text.
>
> **R3-Q2:** The training data compatible of this architecture are limited, which reduces the generalization of the proposed method.
>
>
> 1. Our time-consistency training is specifically designed for deblurring, where constructing blur trajectories is straightforward. The type of dataset required for our training is **not inherently hard to build**. As discussed around Eq. (4), blur–clean pairs can be generated via blur-kernel convolution, consecutive-frame averaging, or real capture. The first two are already widely used in practice due to their low cost and easy alignment. We have already demonstrated that consecutive-frame averaging naturally yields training samples along a blur trajectory (i.e., enlarged GoPro). For blur-kernel convolution, one can simply sample different segments of a generated kernel trajectory to obtain multiple blur levels. For real capture, similar trajectories can be obtained by using multiple cameras with different exposures or a single camera that records images under varying exposure times.
>
> 2. We agree that the training data currently used are limited in diversity, but this is primarily due to **comparison fairness**, not a restriction of our architecture. The motion deblurring community has long relied on the standard setting of *training only on GoPro and evaluating on GoPro/HIDE/RealBlur*, and there is a lack of updated, higher-quality benchmarks. To ensure a fair and directly comparable evaluation, we strictly follow this established protocol.
>
> 3. Importantly, our method only assumes access to a **blur level / timestep** annotation, which can be defined in all three data-generation regimes above, making the approach readily extendable beyond GoPro. Moreover, our results on RealBlur and other test sets show that, even when trained under this “limited” data protocol, FideDiff still exhibits strong real-world generalization in terms of LPIPS/DISTS/FID. This empirically supports that our architecture is not inherently tied to a narrow data distribution and can generalize well in practice.
>
> **R3-Q3:** How to concatenate latent feature z with the estimated blur kernel?
>
> To better exploit the blur kernel, we adopt a filter-like module, as detailed in Section 4.2. The estimated blur kernel $k _{t}$ is first downsampled to $k _{in}$ by a convolution with a stride 8 so that its spatial resolution matches that of the latent feature z. We then concatenate $k _{in}$ and z along the channel dimension, and compute attention weights and subsequent operations on the concatenated feature. In this way, the kernel information is further processed and injected into the latent feature before being fed into the ControlNet branch. The resulting ControlNet features are finally fused back into the U-Net by a zero-initialized convolution added to the skip-connection features, following the standard ControlNet design.
>
> **R3-Q4:** What is the blue block in Figure 4?
>
> In Figure 4, the large **light-blue** block at the bottom represents our foundation model, which includes the SD encoder, UNet backbone, SD decoder, and the discriminator. In the Kernel ControlNet part, the **blue block** at the top right denotes the regression model used in FideDiff for timestep prediction; its detailed architecture (AdaptiveAvgPool2d, Linear, ReLU, Linear, Sigmoid) is described in Section 4 of the supplementary material. The **blue block** at the bottom left of Kernel ControlNet corresponds to $k _{in}$, and the orange block to $z _{in_2}$; these two are concatenated to calculate the weight map. We add corresponding annotations in the revised version of the paper.

---

> ### Author Response · Authors · 2025-11-21
> **Response to Reviewer Hp3A (denoted as R3) part2**
>
> **R3-Q5:** Why can't the model be trained end-to-end instead of using three stages.
>
> We adopt a three-stage training scheme rather than a single end-to-end optimization for both practical and methodological reasons.
>
> First, after finetuning the Foundation Model with LoRA, we merge the LoRA weights into the UNet backbone. During ControlNet training, we follow the standard ControlNet practice: **freeze the UNet parameters** and use these updated weights to **initialize** the ControlNet branch, which stabilizes training and significantly reduces memory and compute cost. Jointly updating the Foundation Model and  ControlNet from the beginning leads to less stable optimization.
>
> Second, this decoupled design makes the system **modular and reusable**. Our Foundation Model serves as a strong, well-defined baseline; future work can plug in alternative conditioning or kernel-injection mechanisms on top of it without retraining the entire diffusion backbone from scratch.
>
> Third, the kernel estimation module is learned from scratch, while Kernel ControlNet is initialized from the pretrained UNet weights. Training them together from random and pretrained initializations, respectively, tends to hurt convergence. We therefore first pretrain the kernel estimation network with the reblur loss $\mathcal{L} _{reblur}$, and then optimize it jointly with Kernel ControlNet in the third stage. Note that stages 2 and 3 are **not completely disjoint**: $\mathcal{L} _{reblur}$ is still used in stage 3 (with $\lambda _3 = 0.1$) as a regularizer for kernel estimation.
>
> We will clarify these design choices and their empirical motivation in the revised manuscript.
>
> **R3-Q6:** evaluation of time prediction
>
> Below, we present the timestep prediction results on GoPro and HIDE. For these images, the ground-truth timestep is $t_{GT} = 200$. We observe that the predictions of the timestep module are mostly concentrated around 200, with slight biases differing across the two datasets.
>
> | timestep | GoPro | HIDE |
> |----------|-------| -----|
> | <= 79  | 0 | 17 |
> | 80~119 | 0 | 116 |
> |120~159 | 11 | 319 |
> | 160~199| 83 | 1009 |
> |200~239 | 740 | 538 |
> |240~279 | 196 | 26 |
> |>= 280  | 81 | 0 |
>
> **R3-Q7:** The data generation process.
>
> Our intention is as follows. The original GoPro dataset is captured at 240 fps, where each blurry (LQ) image is obtained by averaging 7 or 11 or 13 consecutive frames, and the middle frame is used as the corresponding sharp (HQ) image. To enlarge GoPro while preserving all original samples, we proceed as follows:
>
> For sequences where the original LQ is generated by averaging 13 frames, we keep the same central HQ frame and construct additional blur levels by changing the averaging window. Specifically, we average 15 frames (7 before + center + 7 after) and 11 frames (5 + 1 + 5) around the same HQ. In this way, each HQ is associated with three LQ images of different blur severity (11-, 13-, and 15-frame averages).
>
> For sequences where the original LQ is generated by averaging 11 frames (which constitutes the majority of GoPro), we further densify the sampling along the same blur trajectory via probabilistic augmentation. Concretely, we generate additional blurry images by averaging different numbers of frames (5, 7, 9, 13, and 15) around the same HQ with a prescribed probability schedule. As a result, for an original 11-frame LQ–HQ pair, we obtain an expanded set of averages with an approximate ratio of 5:7:9:11:13:15 = 0.4:0.45:0.45:1:1:0.7.
>
> We will clarify this data generation process more explicitly in the revised supplementary material and hope this resolves your confusion.

---

> > ### Comment · Reviewer_Hp3A · 2025-11-27
> >
> > Thanks authors for their clarification. I am satisfied with the response.

---

> > > ### Author Response · Authors · 2025-11-27
> > >
> > > Dear Reviewer Hp3A,
> > >
> > > We are delighted to see that our response addressed your concerns. Thank you for acknowledging the value of our work and helping us improve the quality of our manuscript!
> > >
> > > Best regards,
> > >
> > > Authors

---

### Official Review · Reviewer_qJtp · 2025-10-28

**Soundness:** 3
**Presentation:** 3
**Contribution:** 3
**Rating:** 6
**Confidence:** 3

**Summary:**

The paper proposes FideDiff, a single-step diffusion model that reformulates motion deblurring as a diffusion-like process where each timestep corresponds to a progressively blurred image rather than Gaussian noise levels.
The core technical contributions include a consistency training paradigm where the model learns to map all blur levels along a trajectory to the same clean image and Kernel ControlNet, which integrates blur kernel estimation into the diffusion model via a filter-like module
The model is built upon Stable Diffusion 2.1 and evaluated on GoPro, HIDE, and RealBlur datasets.

**Strengths:**

1. **Problem Formulation Is Reasonable**: The paper clearly discusses the limitations of existing diffusion-based deblurring methods (significant inference time, fidelity-perception tradeoff) and provides concrete evidence (Figure 2) showing the tradeoff between steps and fidelity in super-resolution as motivation.

2. **Consistency Training Framework**: The reformulation of deblurring as a diffusion-like process where timesteps correspond to blur severity rather than noise levels is reasonable. The insight that temporal consistency can enable single-step inference is valuable.

3. **Comprehensive Experimental Evaluation**: The paper includes extensive comparisons across multiple datasets (GoPro, HIDE, RealBlur-J, RealBlur-R) and metrics (PSNR, SSIM, LPIPS, DISTS), with both transformer-based and diffusion-based baselines. Tables 4-7 and Figures 7-8 systematically evaluate the contribution of each component (foundation model design choices, Kernel ControlNet variants, consistency training, timestep prediction).

**Weaknesses:**

1. **Conceptual Novelty and Practical Advantages** Single-step diffusion with Kernel ControlNet and GAN discriminator is conceptually similar to transformer-based models with discriminators. The main distinction is initialization from pre-trained Stable Diffusion weights. However, this does not translate to practical advantages, the performance (both speed and metrics on academic benchmarks) is still a bit inferior or similar to transformer-based methods

2. **Dataset Scale**
The training relies on the enlarged GoPro dataset containing only ~7,877 image pairs, which might be too small to efficiently fine-tune large-scale generative models, which partially can explain the performance gap.

Minor:
- The filter-like module F (Section 4.2) is presented as novel, but it's a relatively standard way to inject spatial information
- The improvement over vanilla ControlNet is small: 0.06 PSNR (28.73 → 28.79) showing that theoretically correct kernel fusion doesnt have signifact impact on model performance.
- Table 5 also shows "motion align" module performs similarly (28.75 PSNR), suggesting the specific design choice is not critical
- Line 013: "true-world modeling" should be "real-world modeling"

**Questions:**

- What is the actual benefit of using pre-trained diffusion models if both performance and speed are inferior to specialized transformer models? This brings an interesting discussion. The authors claim that generative models produce higher image quality due ot their ability to generate/hallucinate scene details. Yet in most examples in the supplementary, the result is nearly identical across all state-of-the-art models. In addition, the evaluation still focuses on standard image similarity metrics: PSNR / SSIM / LPIPS, favoring noisier solutions that hallucinate less (transformer-based models). To demonstrate the effectiveness of the method, authors should consider computing image quality metrics (FID and others) or conducting a user preference study.
- Do you think the current approach might require orders of magnitude more data than transformer methods to achieve superior performance to transformer-based methods?

---

> ### Author Response · Authors · 2025-11-21
> **Response to Reviewer qJtp (denoted as R2) part1**
>
> **R2-Q1:** Dataset Scale.
>
> Thank you for your recognition of our work! Let us first address your questions regarding the dataset, as this will help us discuss the subsequent issues.
>
> Firstly, we acknowledge your observation that 7,877 is indeed insufficient for fine-tuning a Stable Diffusion model. However, in practice, we followed prior work by cropping 1280x720 images into 512x512 patches, resulting in a total of **63k** patches, which, in terms of quantity, is comparable to the datasets (80k~100k patches) used in previous work to fine-tune SD for real-world super-resolution tasks. We used this dataset because the academic community has relied on this fixed setting for motion deblurring research, with few updated, high-quality datasets. To ensure **fair comparison**, we followed the same experimental setup: training only on GoPro and evaluating on other datasets.
>
>
> That being said, for diffusion models to surpass transformer-based models, they require **model improvements** (e.g., foundation models, Kernel ControlNet, and time consistency training in our work) and **1) more diverse and 2) higher-quality datasets**. The GoPro dataset is neither high-quality nor diverse (with limited scenes and few images). To demonstrate the impact of more data, we combine the GoPro, HIDE, and RealBlur datasets, retrain FideDiff and AdaRevD (denoted FideDiff* and AdaRevD*), and evaluate them on four test sets. The results in the table below, show significant performance gains for FideDiff on HIDE, RealBlur-J, and RealBlur-R, with further advancements in LPIPS/DISTS over AdaRevD on RealBlur-J and RealBlur-R. In contrast, AdaRevD degrades on both GoPro and HIDE and does not learn the motion blur patterns of RealBlur as well as FideDiff. This highlights the **potential of diffusion models**—more data indeed enhances performance, while smaller transformer-based models may not achieve the same improvements.
>
>
> | Dataset | Metrics | AdaRevD | AdaRevD* | FideDiff | FideDiff* |
> | --- | --- | --- | --- | --- | --- |
> | GoPro | PSNR↑ | **34.60** | 34.13 | **28.79** | 28.67 |
> |       | SSIM↑ | **0.9716** | 0.9688 | **0.9148** | 0.9136 |
> |       | LPIPS↓ | **0.0712** | 0.0767 | **0.0831** | 0.0857 |
> |       | DISTS↓ | **0.0672** | 0.0681 | **0.0525** | 0.0539 |
> | HIDE | PSNR↑ | **32.35**  | 32.16 | 27.28 | **27.36** |
> |       | SSIM↑ | **0.9525** | 0.9506 | 0.8775 | **0.8820** |
> |       | LPIPS↓ | **0.0872**  | 0.0891 | 0.1068 | **0.0906** |
> |       | DISTS↓ | **0.0666** | 0.0682 | 0.0647 | **0.0584** |
> | RealBlur_J | PSNR↑ | 30.12 | **32.19** | 28.96 | **30.56** |
> |       | SSIM↑ | 0.8945 | **0.9275** | 0.8695 | **0.8811** |
> |       | LPIPS↓ | 0.1408 | **0.1120** | 0.1142 | **0.0696** |
> |       | DISTS↓ | 0.1037 | **0.0967** | 0.0800 | **0.0542** |
> | RealBlur_R | PSNR↑ | 36.53 | **39.82** | 36.01 | **38.11** |
> |       | SSIM↑ | 0.9570 | **0.9727** | 0.9424 | **0.9505** |
> |       | LPIPS↓ | 0.0621 | **0.0461** | 0.0584 | **0.0426** |
> |       | DISTS↓ | 0.0846 | **0.0705** | 0.0862 | **0.0664** |
>
>
> Additionally, we carefully analyze why FideDiff performs adequately on GoPro but excels on RealBlur. We believe that FideDiff cannot significantly outperform AdaRevD on GoPro primarily because the GoPro dataset is of poor quality, built in year 2017. As shown in the table below, we directly test the high-quality images from GoPro and find them to be far inferior to RealBlur (MUSIQ: 47.98–55.75, MANIQA: 0.5668–0.6238). **When forcing a diffusion model, which is designed for high-quality generation, to fit a low-quality dataset, the results are often suboptimal.** On the high-quality RealBlur dataset, however, we achieved SOTA performance, further supporting this point.
>
> | Data | GoPro  | |  |  | RealBlur-J  | | | |
> | --- | --- | --- | --- | --- | --- | --- | --- | --- |
> |  | CLIP-IQA↑ | NIQE↓ | MUSIQ↑ | MANIQA↑ | CLIP-IQA↑ | NIQE↓ | MUSIQ↑ | MANIQA↑ |
> | HQ | 0.2391 | 4.03 | 47.98 | 0.5668 | 0.2622 | 4.13 | 55.75 | 0.6238 |
>
> Therefore, we believe that both the **quantity** and **quality** of the data are critical factors affecting the performance of diffusion models in image restoration. However, to maintain **academic fairness**, we temporarily train only on the GoPro dataset in this work. We will further consider constructing larger and higher-quality datasets to fully unleash the potential of diffusion models.

---

> ### Author Response · Authors · 2025-11-21
> **Response to Reviewer qJtp (denoted as R2) part2**
>
> **R2-Q2:** Conceptual Novelty and Practical Advantages.
>
> | Model | GoPro  | | |  |  | RealBlur-J  | | | | |
> | --- | --- | --- | --- | --- | --- | --- | --- | --- | --- | --- |
> |  | **FID↓** | CLIP-IQA↑ | NIQE↓ | MUSIQ↑ | MANIQA↑ | **FID↓** | CLIP-IQA↑ | NIQE↓ | MUSIQ↑ | MANIQA↑ |
> | Restormer | 8.90 | **0.2557** | 5.18 | 44.96 | 0.5269 | 24.05 | 0.2323 | 5.23 | 48.43 | 0.5639 |
> | AdaRevD | ***6.45*** | ***0.2600*** | 5.03 | ***45.79*** | ***0.5440*** | **20.23** | **0.2469** | 5.22 | **51.34** | **0.5802** |
> |OSEDiff | 24.02 | 0.2120 | 4.52 | 39.28 | 0.4826 | 33.66 | 0.2177 | 5.10 | 41.09 | 0.5105 |
> | Diff-Plugin | 31.80 | 0.2277 | ***4.13*** | 39.01 | 0.4701 | 37.48 | 0.2397 | ***4.59*** | 48.44 | 0.5049 |
> | **FideDiff** | **8.62** | 0.2121 | **4.42** | **45.57** | **0.5437** | ***18.07*** | ***0.2478*** | **4.62** | ***52.74*** | ***0.5883*** |
>
> We add FID and other perceptual quality metrics (no-reference metrics) for comparison. The best metric is ***bold and italicized***, the second best is **bold**.
> In fact, by using pre-trained weights combined with our powerful foundation model, consistency training, and Kernel ControlNet, we have demonstrated practical advantages. These are primarily reflected in:
>
> 1. **Better Perceptual Similarity and Exceptional Performance on Real-World Datasets**: Our model achieves nearly the best results (in FID, LPIPS, DISTS, and other no-reference metrics) on the RealBlur dataset, confirming that the diffusion model, due to extensive pre-training, possesses stronger **real-world** modeling capabilities and **knowledge priors**. This results in better **generalization** and superior performance on real-world datasets compared to other methods.
>
> 2. **Larger Model Capacity and Greater Performance Potential on Diverse Datasets**: While our model may appear less competitive than transformer-based methods in certain areas due to objective factors such as dataset limitations (as mentioned in R2-Q1), diffusion models have far greater potential and scalability due to its strong generation ability. This is a temporary limitation that will improve with future developments.
>
> 3. **Core Contribution and Innovation**: The core contribution of our model lies in how we apply diffusion components like Kernel ControlNet, **time-consistency training**, and timestep prediction. Additionally, the GAN discriminator is performed in the latent space with robust initialization. These innovations enable more accurate and high-fidelity restoration, especially for real-world data.
>
> 4. **Improved Speed at Higher Resolutions**: Our model significantly outperforms multi-step diffusion models in speed, matching the performance of transformer-based methods. While pre-upsampling is used for low-quality datasets (particularly GoPro), it is unnecessary for high-resolution 4K images, allowing our model to regain speed similar to d=8, outperforming transformer-based methods. In **Table 5 of the supplementary materials**, we present memory usage and inference speed for all models, showing that our model maintains superior speed and GPU memory efficiency at higher resolutions, while transformer-based and multi-step diffusion models suffer from increased memory usage and slower speeds.
>
>
>
> We emphasize that our model is the **first to attempt pre-trained single-step diffusion in image motion deblurring**, outperforming all pretrained diffusion-based methods, with excellent real-world generalization. We do not stop at just a strong initialization; we further leverage the potential of diffusion models to enhance fidelity, which provides a valuable direction for the academic community. This work also represents an extension of Stable Diffusion for real-world image restoration, with significant industrial application value. Our model can be integrated with various LoRA adapters to handle different restoration tasks, building a MoE-like model.

---

> ### Author Response · Authors · 2025-11-21
> **Response to Reviewer qJtp (denoted as R2) part3**
>
> **R2-Q3:** filter-like module is a relatively standard way to inject spatial information.
>
> We agree that the concept of using filter-like modules to inject spatial information is not entirely novel. However, this is the first time that we use this technique to inject kernel information into Stable Diffusion (SD) through ControlNet. This approach not only enhances the guidance provided by the kernel information but also extends the usage of ControlNet by allowing it to receive and process kernel information in a more sophisticated manner, unlike traditional methods that simply concatenate kernel data. This novel application of kernel injection into ControlNet distinguishes our approach and expands the capabilities of ControlNet, making it more effective for motion deblurring tasks.
>
> **R2-Q4:** The small improvement over vanilla ControlNet and similar performance of the "motion align" module suggest that the design choices have limited impact on performance.
>
> Apart from the results in Table 5, we add the FID metrics as follows:
> | Model | base | controlnet | motion align | kernel |
> |-------|-------|------------|-------------|--------|
> | GoPro | 8.83 | 8.81 | 8.79 | **8.62** |
> | HIDE |  9.98 | 9.77 | 9.82 | **9.58** |
>
> 1. Kernel ControlNet reaches the best LPIPS/DISTS/FID with a larger margin than PSNR/SSIM. We believe that evaluating pretrained diffusion-based models solely using PSNR and SSIM may be somewhat limited. Metrics like LPIPS, DISTS, and FID, which assess perceptual similarity and distribution alignment, offer a more appropriate evaluation for diffusion models. This is because diffusion models inherently prioritize overall perceptual quality and are less sensitive to small pixel-level errors (such as those captured by PSNR and SSIM).
>
> 2. Kernel estimation is indeed not the only possible approach, but it is currently the most effective methods we have explored. We have established a strong foundation model (which is also one of our key contributions) and introduced an effective approach with Kernel ControlNet. Moving forward, we will continue to explore additional methods for effectively injecting blur information, and we look forward to collaborating with other researchers to further advance this area.
>
>
> **R2-Q5:** Line 013: "true-world modeling" should be "real-world modeling"
>
> Thanks! We have corrected this in the main text.
>
>
> **R2-Q6:** actual benefit of using pre-trained diffusion models?
>
> Please refer to **R2-Q2**, where we show more image quality metrics CLIP-IQA/NIQE/MUSIQ/MANIQA and FID, and discuss about the actual and potential benefits of the pretrained diffusion model.
>
> In the supplementary material, our model shows clearly superior performance on the RealBlur-J dataset compared with both transformer-based and diffusion-based methods, for example on the grass and column contours in scene002-2 and the left-side pattern in 155-18. On the GoPro and HIDE datasets, our model also recovers finer details than diffusion-based baselines in most cases, such as the window structures in GoPro 0869-11-00-000057. Our reconstructions are visually almost identical to transformer-based methods but sometimes more close to the ground truth, e.g., in HIDE 100fromGOPR1089, where the shoe appears slightly clearer in our result.
>
> We also perform user studies under two settings. 20 participants are presented with 30 samples from 4 datasets and asked to select the best-restored image, defined as closest to the ground truth. In the first setting, comparing all diffusion-based models, FideDiff wins 71.11%. In the second, a direct comparison against AdaRevD, FideDiff wins 54.72%.
>
> In summary, even if specialized transformers remain ahead on some PSNR/SSIM numbers on GoPro/HIDE, our experiments show that pretrained diffusion models, when properly adapted as in FideDiff, (1) set a new SOTA among diffusion-based methods, (2) exhibit stronger real-world generalization and perceptual fidelity than transformer-based models and show great potential for future research, and (3) provide a task-reusable, industrially practical backbone for image restoration, rather than merely serving as a slower and weaker alternative to transformers.
>
> **R2-Q7:** more data than transformer methods to achieve superior performance?
>
> Please refer to **R2-Q1**. Overall, we believe that data quality is more critical, while data quantity is also important. Under limited data, higher-quality datasets allow diffusion models to better demonstrate their advantages over transformers (e.g., FideDiff achieves SOTA LPIPS/DISTS/FID on RealBlur), thanks to the rich pretrained priors. At the same time, as the amount of data increases, the potential of diffusion models becomes more evident, whereas transformer-based methods tend to struggle due to their limited model capacity.

---

> > ### Comment · Reviewer_qJtp · 2025-11-25
> >
> > Thanks for the rebuttal! Most of the concerns were addressed. I do agree with the authors that PSNR is not the optimal metric to evaluate the visual quality of the generated content, so the main evaluation should be focused on perceptual quality metrics added in the rebuttal. I would also encourage authors to expand this analysis and provide more extreme visual examples in the supplementary material with severe real-world blur where transformer methods fail, as this might help to demonstrate the fundamental difference between feedforward and diffusion models.
> > My only concern is more conceptual: given a limited amount of available high-quality training data and challenges of scaling up the datasets, I am not convinced that generation models in theory can outperform the feedforward transformer-based solutions. Yet, I believe the paper brings an interesting alternative solutions to existing SOTA methods so I will keep my positive score.

---

> > > ### Author Response · Authors · 2025-11-27
> > >
> > > Dear Reviewer qJtp,
> > >
> > > We really appreciate your feedback and are glad to hear that our response addressed most of your concerns. We are particularly encouraged by your agreement regarding the importance of perceptual quality metrics over PSNR for this task.
> > >
> > > Regarding the visual examples: We think your suggestion to analyze "extreme real-world blur" is excellent. We will include a comprehensive analysis in the supplementary material regarding the suitability of various metrics for deblurring tasks. This will cover pixel-level similarity (PSNR/SSIM), deep perceptual similarity (LPIPS/DISTS), and no-reference overall perceptual quality (CLIP-IQA/MUSIQ/MANIQA). As requested, we will also include a new section in the final version's supplementary material showcasing comparisons on severely blurred images.
> > >
> > > Regarding the conceptual concern on data scaling: We acknowledge your valid point about the challenges of high-quality data scaling. While feedforward models are indeed powerful, we believe the core advantage of the generative approach lies in its ability to leverage learned priors to reconstruct the natural image manifold, especially when the input information is severely degraded (ill-posed problems). We hope our work provides a solid step forward in exploring this alternative path. In our future work, we plan to construct high-quality, realistic blur datasets to further demonstrate the potential and superiority of diffusion models.
> > >
> > > Thank you again for recognizing our work and helping us improve the quality of our paper!
> > >
> > > Best regards,
> > >
> > > Authors

---

### Official Review · Reviewer_jEid · 2025-10-31

**Soundness:** 2
**Presentation:** 3
**Contribution:** 3
**Rating:** 4
**Confidence:** 3

**Summary:**

The paper proposes applying a diffusion-model to the single-image deblurring task. The authors claim that by diffusion process, they achieve high-fidelity sharp image restoration on datasets such as GoPro and RealBlur‑R.

**Strengths:**

The idea of using a diffusion-model for deblurring is interesting in the deblurring community.

The paper presents a comprehensive experimental evaluation with multiple datasets, which shows the authors’ effort to validate the method across settings.

The writing is clearly structured, and the method is described with reasonably good clarity in terms of pipeline, training and inference details.

**Weaknesses:**

1. Numbers inconsistent with public SOTA.

On GoPro, the paper reports PSNR 28.79 dB, far below recent results, e.g., 34.09 dB LoFormer; 34.60 dB AdaRevD. On RealBlur-R, it reports 36.01 dB, while widely reported numbers are 38.6 dB. This raises concerns about evaluation setup and fairness.

2.Perceptual evaluation is insufficient.

For a diffusion model, perceptual quality should be central. The paper underweights CLIP-IQA/LPIPS/DISTS and provides limited qualitative/video evidence. No user study or side-by-side videos under realistic motion is shown.

3. Cost–benefit trade-off is unclear.

The method adds diffusion + kernel control modules, yet runtime, memory, and step count are not thoroughly compared to strong CNN/Transformer baselines under the same hardware. Efficiency claims are therefore hard to judge.

4. Ablations not fully isolating gains.

The paper does not cleanly separate the contributions of the foundation diffusion, Kernel ControlNet, and t-prediction across datasets. Stronger ablations and cross-dataset tests are needed to establish robustness.

**Questions:**

Could you provide more visual materials such as videos?

---

> ### Author Response · Authors · 2025-11-21
> **Response to Reviewer jEid (denoted as R1) part1**
>
> **R1-Q1:** Numbers inconsistent with public SOTA.
>
> Our experiments are conducted with **fair and consistent** settings, following the established practices in the deblurring domain. All models are trained exclusively on the GoPro training set and tested on GoPro, HIDE, and RealBlur datasets, with identical metric implementations across all models. The numerical results of AdaRevD in our paper are consistent with those reported in the original paper. Regarding the 38.6 dB PSNR on RealBlur-R mentioned in your comment, we have not yet found a relevant paper reporting this result. However, in other settings (where models are trained and tested directly on RealBlur), such high PSNR values can indeed be achieved.
>
> It is important to note that for all pretrained diffusion-based methods, FideDiff achieves SOTA performance across all metrics. For well-developed transformer-based methods, such as AdaRevD, FideDiff achieves SOTA in at least one metric (LPIPS/DISTS) across all datasets, with both metrics reaching SOTA on RealBlur-J. Additionally, PSNR and SSIM on RealBlur-J/R are also comparable to transformer-based methods.
>
> It is worth emphasizing that transformer-based methods excel at **pixel-level** modeling, whereas diffusion models are better suited for overall **perceptual consistency** and exhibit stronger real-world modeling and generalization capabilities. Even though there is a gap in PSNR, the human eye might not perceive noticeable visual differences.
>
> **R1-Q2:** Perceptual evaluation is insufficient.
>
> Thank you for the suggestion regarding further evaluation! Firstly, it is important to note that our paper focuses on taming the diffusion model for **high-fidelity** image restoration (deblurring), prioritizing similarity to the ground truth (GT) clean image, as measured by full-reference metrics(PSNR/SSIM/LPIPS/DISTS), and not optimizing the no-reference metrics(CLIPIQA/MUSIQ), as the overall perceptual quality alone is not suitable for the restoration task.
>
> Indeed, we place significant emphasis on **LPIPS/DISTS** metrics throughout our paper, as they are particularly effective at evaluating the perceptual similarity between the restored image and the GT image, as shown in Tables 2 and 4-7. As a pretrained-diffusion-based model, FideDiff achieves excellent performance across all datasets on these perceptual metrics. We will further clarify this in our revision.
>
> Additionally, we also include widely used perceptual quality metrics such as CLIP-IQA, NIQE, MUSIQ, and MANIQA among the diffusion models, as detailed below:
> | Model | GoPro  | |  |  | RealBlur-J  | | | |
> | --- | --- | --- | --- | --- | --- | --- | --- | --- |
> |  | CLIP-IQA↑ | NIQE↓ | MUSIQ↑ | MANIQA↑ | CLIP-IQA↑ | NIQE↓ | MUSIQ↑ | MANIQA↑ |
> |OSEDiff | 0.2120 | 4.52 | 39.28 | 0.4826 | 0.2177 | 5.10 | 41.09 | 0.5105 |
> | Diff-Plugin | **0.2277** | **4.13** | 39.01 | 0.4701 | 0.2397 | **4.59** | 48.44 | 0.5049 |
> | **FideDiff** | 0.2121 | 4.42 | **45.57** | **0.5437** | **0.2478** | 4.62 | **52.74** | **0.5883** |
>
> The best metric is **bold**. FideDiff achieves the best performance on most perceptual metrics compared to other diffusion-based models.
>
> To further validate restoration quality, we perform a user study. 20 participants are presented with 30 samples from 4 datasets and asked to select the best-restored image, defined as closest to the ground truth. Comparing all diffusion-based models, FideDiff wins **71.11%**.
>
> For video comparisons, we first clarify that although the GoPro dataset is derived from consecutive video frames, all models target image motion deblurring, which is a distinct task from video deblurring. For instance, the RealBlur dataset lacks inter-image scene relationships and cannot form videos. Nevertheless, we manually synthesized videos from GoPro restoration results at 10FPS. **Please refer to the supplementary materials for 2 scene samples**, synthesized from 200 images. FideDiff's deblurred results significantly outperform OSEDiff and Diff-Plugin, closer to the high-quality (GT) videos.

---

> ### Author Response · Authors · 2025-11-21
> **Response to Reviewer jEid (denoted as R1) part2**
>
> **R1-Q3:** Cost–benefit trade-off is unclear.
>
> In our experiments, we strictly and fairly test the inference speed on A800 GPUs, as shown in **Table 3** of the main text. We demonstrate the running speed of our model without pre-upsampling and without adding Kernel ControlNet (the last three rows). Moreover, our model is **single-step** throughout the paper. In Table 3, all diffusion models are labeled with "-s'n'", indicating the number of steps used for inference. In **Table 5 of our updated supplementary materials**, we include the memory usage and inference speed of all models during inference and provide further analysis of the input resolution. Our model demonstrates a more significant advantage in speed and GPU memory usage when increasing image resolution, whereas transformer-based and multi-step diffusion-based models require substantial GPU memory and experience a significant slowdown in inference speed.
>
>
>
> **R1-Q4:** Ablations not fully isolating gains.
>
> Thank you for your suggestions! Our contributions are indeed numerous, including the foundation model, an enlarged GoPro dataset, consistency training (CT), Kernel ControlNet (KCN), and Time-Prediction (TP). In order to clearly illustrate the contribution of each part, we create the table below for your reference. Each row in this table corresponds to a specific experiment from the original paper. Please refer to the positions indicated in the last column for the corresponding results in our original paper. We will also consider restructuring the paper or updating the supplementary materials to further elaborate on the contribution of each part.
> | Enlarged GoPro | CT | KCN | TP | PSNR↑ | SSIM↑ | LPIPS↓ | DISTS↓ | Position |
> |----|-----|-----|-----|----| ----|-----|-----|----|
> |   |   |   |   | 28.51 | 0.9101 | 0.0899 | 0.0555 | Table 4 (main) |
> |   |   | ✓ |   | 28.56 | 0.9109 | 0.0873 | 0.0549 | Table 7 (main) |
> | ✓  |   |   |   | 28.65 | 0.9130 | 0.0892 | 0.0545 |Table 4 (supplementary)|
> | ✓   | ✓   |   |   |28.68 | 0.9130 | 0.0854|  0.0533 | Table 5 (main) |
> | ✓  |   |  ✓ |   |28.74 | 0.9142 |  0.0871 | 0.0548 | Table 6 (main) |
> | ✓  |  ✓ |  ✓ |  ✓  | 28.62 | 0.9123 | 0.0828 |  0.0522 | Table 6 (main) |
> | ✓  |  ✓ |  ✓ |   | 28.79 | 0.9148 | 0.0831 | 0.0525 | Table 5/6/7 (main) |
>
> For the enlarged GoPro, CT, and TP, we have performed cross-validation on all four datasets (Table 6/7). For KCN, we also conducted cross-validation on GoPro and HIDE (Table 5); here we did not select RealBlur because it lacks a definitive timestep $t$ as the ground truth, which impacts the ablation study of KCN. As for our Foundation model, since it involves basic loss functions and parameter settings, we only conducted experiments on GoPro, but we have updated the corresponding cross-validation results in Table 2 in the supplementary materials of our Foundation model.

---

> > ### Comment · Reviewer_jEid · 2025-11-26
> >
> > That addressed my concerns. I've improved to a score of 6.
> >
> > Additionally, some references are missing publication information. Please fix this in future.
> >
> > Junhao Cheng, Wei-Ting Chen, Xi Lu, and Ming-Hsuan Yang. Unpaired deblurring via decoupled diffusion model, 2025.
> >
> > Jue Gong, Jingkai Wang, Zheng Chen, Xing Liu, Hong Gu, Yulun Zhang, and Xiaokang Yang. Human body restoration with one-step diffusion model and a new benchmark, 2025.

---

> > > ### Author Response · Authors · 2025-11-27
> > >
> > > Dear Reviewer jEid,
> > >
> > > Thank you very much for your time and for improving the score! We are glad that our clarifications were helpful.
> > > We have noted the missing information in the references you mentioned. We will strictly update the citations for these references and ensure all references are complete in the final version of the paper.
> > >
> > > Best regards,
> > >
> > > Authors

---

### Author Response · Authors · 2025-11-21
**Response to all reviewers and area chairs for a brief summary**

Dear reviewers and area chairs,

We sincerely thank all reviewers and area chairs for the insightful feedback.

We are pleased to receive the recognition that:

1. Reviewers jEid and n57E acknowledge the novelty of using a diffusion model for deblurring, especially reviewer n57E, who thinks our work encourages other researchers to explore diffusion-based, high-fidelity methods for other restoration tasks.

2. Reviewers qJtp and Hp3A praise the reasonable formulation of deblurring as a diffusion-like process, which is modeled by solid mathematical analysis and improves robustness to severe blurs.

3. Both Reviewer Hp3A and n57E note the high-fidelity results surpassing all diffusion-based methods. Reviewers qJtp and Hp3A emphasize the significance of time-consistency training and the proper injection of the kernel information.

4. All reviewers commend the clear structure and well-organized writing of the paper, and appreciate the thorough and comprehensive experimental evaluation, with systematic comparisons to baselines and a clear assessment of each method component (Tables 4-7, Figures 7-8).

We have responded to each reviewer individually and would like to summarize our responses here:

1. We modify ambiguous sections in the main paper to clarify the expression and the equations, and revise the details in Figure 4.

2. We analyze the advantages and potential of applying diffusion models to low-level vision tasks, particularly deblurring.

3. We analyze the issues related to the training data and explore the potential for further performance improvements.

4. We incorporate no-reference image quality metrics and add a user study to demonstrate the superiority of FideDiff.

5. We conduct additional ablation studies on our model, including exploring the learnable text embedding, etc.

We extend our gratitude again to all reviewers and area chairs!

Best regards,

Authors

---

### Author Response · Authors · 2025-11-29
**Summary of Rebuttal Progress and Key Clarifications**

**Dear Program Chairs, Senior Area Chairs, Area Chairs, and Reviewers,**

We sincerely thank all program chairs, senior area chairs, area chairs, and reviewers (**R1-jEid, R2-qJtp, R3-Hp3A, R4-n57E**) for your time and dedication to ensuring a fair review process. We understand the significant additional workload caused by the recent OpenReview incident. To assist in your reassessment, we would like to provide a concise summary of our paper’s status and the consensus reached:

**1. Rebuttal Progress (Pre-Incident)**

Before the score reversion, we engaged in active discussions with **all four reviewers**.
* **Initial Scores:** 8, 6, 6, 4 (Conf. 4, 4, 3, 3)
* **Post-Rebuttal Scores (as of Nov 27):** **8, 8, 6, 6 (Conf. 4, 4, 3, 3)** — *Explicit confirmation of the score raises is documented in the "Official Comments" by Reviewers jEid and n57E.*
* **Status:** All reviewers responded positively to our rebuttal, and the score increases were based on technical clarifications provided before the leak was publicized.

**2. Reviewer Consensus on Strengths**

There is a strong consensus among reviewers (jEid, qJtp, Hp3A, n57E) regarding the value of our work:
* **Novelty & Impact:** Reviewers acknowledge the novelty of our diffusion-based framework and its potential to inspire future high-fidelity restoration research.
* **Solid Formulation:** Reviewers praise our diffusion-like formulation, emphasizing that the solid mathematical analysis significantly improves robustness to severe blurs.
* **Superior Performance:** Reviewers highlight that our results surpass all existing diffusion-based methods, validating the effectiveness of time-consistency training and kernel injection.
* **Quality & Rigor:** All reviewers commend the clear writing and comprehensive experimental evaluation (Tables 4-7, Figures 7-8).

**3. The Concerns We Addressed During Rebuttal**

In response to reviewer feedback, we provided comprehensive clarifications accompanied by new experiments and analyses:
* **Enhanced Evaluation:** We provided additional numerical results, extensive visual comparisons, and a user study, conclusively demonstrating FideDiff's superior performance in overall image quality.
* **In-depth Analysis:** We conducted a **comprehensive analysis** regarding comparison fairness and the impact of data quality/quantity, while clarifying our model's specific advantages and potential over Transformer-based approaches.
* **Ablations & Clarifications:** We included more ablation studies of some minor components and conducted additional memory/speed tests across various resolutions. We also revised ambiguous text, mathematical notation, and figures to improve clarity.

### **Overall Summary**

Our work innovatively applies pretrained Diffusion models to image deblurring by rigorously reformulating the forward and reverse processes. We introduced consistency training to support efficient single-step sampling, significantly reducing inference time. We established a powerful foundation model and designed a Kernel ControlNet to inject additional information, demonstrating the model's efficiency and effectiveness through extensive and comprehensive experiments.

Although there were initial discussions regarding datasets and model potential, concerns calling for additional qualitative/quantitative comparisons, and specific questions and suggestions regarding details, we successfully addressed these issues through active engagement. We received positive responses and acknowledgment from all four reviewers, resulting in a score increase from an initial 8/6/6/4 to **8/8/6/6 (Confidence: 4/4/3/3)**.

We hope this summary assists you in navigating the review history efficiently. We remain committed to the scientific integrity of ICLR and thank you again for your efforts in handling our submission.

Sincerely,

**Authors of Submission 2583**

---

### Meta-Review · Area_Chair_tZ7o · 2026-01-07

**Summary:**

This paper received mixed review rating in the pre-rebuttal stage.

The major improvements from the rebuttal period were

- Enhanced Evaluation
- In-depth Analysis
- Ablations & Clarifications

The reviewers were all confirmed with the responses and results provided by the authors.

Eventually, the paper received ratings: 8, 8, 6, 6.

The AC checked the paper again and does agree that the paper has merit of research novelty and theoretical breakthrough that could be beneficial to ICLR readers if accepted.

**Reviewer Concerns:**

See above.

**Reviewer Scores:**

The reviewers actively interacted with the authors.
The major concerns were addressed by the authors.

A consensus among reviewers finally reaches to both novelty and theoretical formulation of the problem, accompanied by solid experiments.

---

### Decision · Program_Chairs · 2026-01-26

Accept (Poster)